# Technical Debt in In-Context Learning: Diminishing Efficiency in Long Context

**Taejong Joo & Diego Klabjan**
Department of Industrial Engineering & Management Sciences
Northwestern University
Evanston, IL, USA
`{taejong.joo,d-klabjan}@northwestern.edu`

## Abstract

Transformers have demonstrated remarkable in-context learning (ICL) capabilities, adapting to new tasks by simply conditioning on demonstrations without parameter updates. Compelling empirical and theoretical evidence suggests that ICL, as a general-purpose learner, could outperform task-specific models. However, it remains unclear to what extent the transformers optimally learn in-context compared to principled learning algorithms. To investigate this, we employ a meta ICL framework in which each prompt defines a distinctive regression task whose target function is drawn from a hierarchical distribution, requiring inference over both the latent model class and task-specific parameters. Within this setup, we benchmark sample complexity of ICL against principled learning algorithms, including the Bayes optimal estimator, under varying performance requirements. Our findings reveal a striking dichotomy: while ICL initially matches the efficiency of a Bayes optimal estimator, its efficiency significantly deteriorates in long context. Through an information-theoretic analysis, we show that the diminishing efficiency is inherent to ICL. These results clarify the trade-offs in adopting ICL as a universal problem solver, motivating a new generation of on-the-fly adaptive methods without the diminishing efficiency. [1]

## 1 Introduction

Transformers, particularly large language models (LLMs), are able to perform *in-context learning* (ICL) [1]; they can adapt to new tasks simply by conditioning on demonstrations in their input prompt [2]. Not only conveniently operated without any explicit parameter updates, but ICL even with just a few demonstrations (a.k.a. *few-shot* ICL) surprisingly outperforms task-specific state-of-the-art models in diverse tasks, from question answering to common sense reasoning [3, 4, 1].

This raises a fundamental question whether ICL can act as a universal learner, replacing task-specific models. To answer this, we must first address a more precise question: *How optimal is ICL as a learning algorithm, compared to principled learning algorithms?* In principle, this could be answered by exhaustively benchmarking ICL against principled learning algorithm across varied data and model scales [5, 6] and task types [7, 8]. However, the computational demands for training modern LLMs pose significant challenges for such direct comparisons. The goal of this work is to answer the question without such prohibitive computational demands.

To answer the question, theoretical studies have analyzed *asymptotic behavior of ICL* using rich tools from statistics and learning theory, such as regret and generalization bounds [9–12]. However, these asymptotic results fall short of fully characterizing real-world LLM behavior. For instance, the regret

---

[1]Our source code is available at `https://github.com/tjoo512/technical-debt-in-icl`.

39th Conference on Neural Information Processing Systems (NeurIPS 2025).

upper bound for LLMs become nearly vacuous in few-shot regimes [13, 14], which cannot explain the striking few-show ICL performances. Moreover, because other principled learning algorithms have the similar asymptotic behavior, it remains unclear whether ICL is a *better* learning algorithm than such learning algorithms.

*Physics-style* or synthetic benchmarking approaches have provided valuable insights that *transformers might optimally learn in-context*, isolating core aspects of LLM training in controlled environments [15–17]. These approaches by nature can enable an efficient, comprehensive comparison between ICL and principled learning algorithms with arbitrarily high levels of statistical significances, providing insights that often generalize to real-world LLMs despite inherent simplifications (see Appendix A.1 for more detailed discussion on the usage of stylized setting). Notably, Garg et al. [16] and follow-up works [18, 19] present various stylized settings where the ICL performances across different demonstration sizes resembles the learning curve of the optimal learning algorithm (e.g., Figure 3a). However, these works have not yet provided an explicit relationship between relevant quantities (e.g., sample complexity and the optimality gap). Thus, the question of *to what extent* transformers can learn optimally in-context remains unanswered.

To quantify optimality of ICL as a learning algorithm, we compare ICL's sample complexity-related measures to those of principled learning algorithms by revisiting the performance profiles [20]—classic benchmarking framework for optimization software. As a result, we uncover a new insight on optimality of ICL in §3: *While ICL with few-shot demonstrations achieves near optimal sample complexity, ICL's sample complexity sharply deteriorates as the number of demonstrations increases in long context.* Concretely, many-shot ICL often requires 1.5 times more demonstrations than the Bayes optimal estimator to achieve the same performance. This indicates that, although transformers are theoretically capable of implementing principled algorithms in-context [19], their in-context learning behavior deviates significantly from the optimal learning algorithm in the many-shot regime. We further present evidence that, unlike principled algorithms, ICL may lack fundamental statistical properties (e.g., consistency and asymptotic efficiency) that are critical for algorithms to effectively learn from large demonstration sizes. Crucially, as ICL performances generally improve with more demonstrations, this novel insight would be difficult to uncover without directly comparing ICL to the principled learning algorithms with proper sample complexity measures.

To solidify this empirical finding, we provide information-theoretic analyses demonstrating that *the diminishing efficiency is intrinsic to the ICL mechanism itself* in §4. Specifically, we prove that ICL without diminishing efficiency has stringent necessary conditions (e.g., negligible excess risk), and the result is independent to particular instantiation of models and environments. The results explain ICL's deficient sample complexity compared to the principled learning algorithm in many-shot regimes.

Taken together, our work unveils a hidden *technical debt* in the ICL mechanism, suggesting a nuanced view of ICL as a universal problem solver: the price we pay for its training-*free* adaptability is a fundamental inefficiency in sample complexity that compounds as we push toward higher performance targets with the current ICL mechanism *as is*. Crucially, this debt appears intrinsic to the ICL mechanism and thus unlikely to be serviced by simply scaling data and model sizes. We hope these insights clarify the trade-offs in adopting ICL as a universal problem solver and motivate a new generation of "on-the-fly" adaptive methods without the diminishing efficiency.

## 2 Setup

In §2.1, we describe the meta ICL environment for evaluating ICL as a learning algorithm, followed by designs of a transformer for solving the meta ICL task (§2.2). We then devise principled predictors (§2.3) and compare them with transformers using performance measures defined in §2.4.

### 2.1 Meta ICL Environment

In the meta ICL [16], each prompt characterizes an instance of a learning problem. Specifically, a prompt $H_T$ consists of demonstrations with a test input, i.e., $H_T \triangleq (X_1, Y_1, \cdots, X_T, Y_T, X_{T+1})$, and each output is generated by some function $f^*$, i.e., $Y_t = f^*(X_t)$ for $t \in [T+1] \triangleq \{1, 2, \cdots, T+1\}$. Here, the goal of a transformer is formalized as accurately predicting $Y_{T+1}$ with $H_t$, which requires to (implicitly) infer the underlying function $f^*$ from the demonstrations. We denote the set of demonstrations as $D_T \triangleq (X_1, Y_1, \cdots, X_T, Y_T)$. Following the meta ICL literature [16, 18,

19, 6], we focus on regression problems where many principled learning algorithms can be derived analytically (cf. §2.3).

For the data generating distribution of a prompt $H_T$, we follow the approach of sampling target functions $f^*$ from a *hierarchical distribution* [21] to capture a more interesting aspect of a learning algorithm—model selection. Under the hierarchical $f^*$, the prompt $H_T$ is realized by the following sampling process, which is denoted as $H_T \sim \mathbb{P}(\cdot; \mathcal{E})$ with parameters $\mathcal{E} \triangleq ([M], \sigma_w^2, \sigma_\epsilon^2)$.

**1)** Sample the implicit dimension $m \sim \mathcal{U}([M])$ from a uniform distribution over set $[M]$ and construct the (unobservable) feature space $\Phi_m(x) \triangleq [1, \cos(\frac{\pi x}{\mathcal{T}}), \sin(\frac{\pi x}{\mathcal{T}}), \cdots, \cos(\frac{m\pi x}{\mathcal{T}}), \sin(\frac{m\pi x}{\mathcal{T}})]$ where $\mathcal{T} > 0$ controls the frequency of the trigonometric functions.

**2)** Sample weight $w_m \sim \mathcal{N}(0, \sigma_w^2 \mathbf{I}_{2m+1})$, where $\mathbf{I}_{2m+1}$ is the identity matrix with rank $2m + 1$. The weight $w_m$ defines the target function $f^*(x) \triangleq w_m^\top \Phi_m(x)/\sqrt{m+1}$ where the constant $\sqrt{m+1}$ makes the variance of $f^*$ remains constant across different $m$. We let $\mathcal{F}_m \triangleq \{w^T \Phi_m(\cdot) | w \in \mathbb{R}^{2m+1}\}$ denote the set of all target functions with implicit dimension $m$.

**3)** Construct a prompt $H_T$ with a test output $y_{T+1}$ by $x_t \sim \mathcal{U}([x_{\min}, x_{\max}]), y_t = f^*(x_t) + \epsilon_t$ for $t \in [T + 1]$, where $\epsilon_t \sim \mathcal{N}(0, \sigma_\epsilon^2)$ is a random observation noise.

This hierarchical sampling involves a rich class of functions since $\{1\} \cup \{\cos(\frac{m\pi x}{\mathcal{T}})\}_{m \in \mathbb{N}} \cup \{\sin(\frac{m\pi x}{\mathcal{T}})\}_{m \in \mathbb{N}}$ forms a basis of square-integrable functions on $[x_{\min}, x_{\max}]$. Following Panwar et al. [21], we set $\mathcal{T} = x_{\max} = -x_{\min} = 5$ and $M = 10$ (our findings are indifferent to these values).

We benchmark ICL with respect to different configurations of $\mathcal{E}$, called *scenario*, to enable comprehensive evaluations that could be encountered in practical scenarios (e.g., low signal-to-noise ratio (SNR), defined as $Var(f^*)/\sigma_\epsilon^2$, for emulating a highly noisy environment). We denote $\mathcal{S}$ as a set of scenarios and $\mathcal{E}_s$ as parameters of a scenario $s \in \mathcal{S}$. We also have $H_T^s \triangleq (X_1^s, Y_1^s, \cdots, X_{T+1}^s)$ generated from $\mathbb{P}(\cdot; \mathcal{E}_s)$ for each scenario $s$, where we omit superscripts when there is no ambiguity.

## 2.2 Transformers

For a transformer $\text{TF}_\theta$, we adopt the setup from Garg et al. [16] and follow-up works [21, 19, 18, 6] that use the GPT-2 [22] architecture (cf. details in Appendix A.2). For optimizing $\theta$ in the pretraining stage, we use the following minimization objective

$$\mathcal{L}(\theta) \triangleq \mathbb{E}_{H_{T_{\text{train}}}} \left[ \frac{1}{T_{\text{train}}} \sum_{t=0}^{T_{\text{train}}-1} (\text{TF}_\theta(H_t) - Y_{t+1})^2 \right] \tag{1}$$

where $H_{T_{\text{train}}}$ is generated by the prompt distribution described in §2.1. We set $T_{\text{train}} = 50$ for all scenarios, which is roughly $2 \cdot (2M + 1)$ as in the previous works [16, 21], and train $\text{TF}_\theta$ separately for each scenario.

## 2.3 Principled Baselines

To benchmark ICL, we derive principled baselines that learn from demonstrations $D_t$ and produce a prediction function $f_b(\cdot; D_t)$, where $b$ is the identifier of a particular baseline. We denote $f_b^t(x) \triangleq f_b(x; D_t)$ and $f_{\text{ICL}}^t(X_{t+1}) \triangleq \text{TF}_\theta(H_t)$ whenever there is no ambiguity.

The optimal baseline is Bayesian model averaging (BMA), which makes prediction by aggregating models from different hypothesis classes. Formally, it is defined as

$$f_{\text{BMA}}^t(x) \triangleq \sum_{m \in [M]} p(\mathcal{F}_m \mid D_t) \, \hat{w}_m^\top(D_t) \Phi_m(x), \tag{2}$$

where $p(\mathcal{F}_m \mid D_t)$ is the posterior probability of $\mathcal{F}_m$ and $\hat{w}_m$ is the ridge regression estimator for $\mathcal{F}_m$, defined as $\hat{w}_m(D_t) = (\Phi_{m,t}^\top \Phi_{m,t} + \frac{\sigma_\epsilon^2}{\sigma_w^2} \mathbf{I}_{2m+1})^{-1} \Phi_{m,t}^\top Y_t$ with $\Phi_{m,t} \in \mathbb{R}^{t \times (2m+1)}$ whose $k$-th row is $\Phi_m^\top(X_k)$ and $Y_t = (Y_1, \cdots, Y_t) \in \mathbb{R}^t$. The Bayes optimal estimator defined in (2) minimizes the expected risk with respect to the true hierarchical data-generating distribution. This is distinct from the notion of optimality in Raventós et al. [6], which is defined with respect to an empirical pretraining distribution with a finite number of samples, such that deviating from it can lead to better generalization. The optimality of BMA follows from two standard results that (1) $\mathbb{E}[Y_{t+1}|H_t]$ is a solution to $\min_{f \in \mathcal{F}} \mathbb{E}_{Y_{t+1}} [l(f(X_{t+1}; D_t), Y_{t+1}) \mid H_t]$ almost everywhere for

all $t \in \mathbb{N}$ and $\mathcal{F}$ being the set of all functions from $H_t$ to $\mathbb{R}$ (e.g., Lemma 1 in [23]) and (2) $\mathbb{E}[Y_{t+1}|H_t] = \sum_{m \in [M]} p(\mathcal{F}_m|D_t)\mathbb{E}[Y_t|\mathcal{F}_m, H_t] = f_{\mathsf{BMA}}^t(x)$ (e.g., Equation 3.58 in [24]).

We also consider a family of principled baselines that embodies different model selection strategies with the same model fitting capacity as the optimal predictor. Such baselines make predictions by

$$ f_b^t(x; D_t) = \hat{w}_{m_b^\dagger}^\top(D_t)\Phi_{m_b^\dagger}(x), \tag{3} $$

where $m_b^\dagger \in \arg\max_{m \in [M]}\{\mathrm{Score}_b(m)\}$ with $\mathrm{Score}_b(\cdot)$ being some model selection criterion of $b$.

### 2.4 Measures for Benchmarking Optimality of ICL

Inspired by seminal work [20] that benchmarks (deterministic) optimization software, we first define the base metric measuring the optimality of a learning algorithm in $s \in \mathcal{S}$. Then, we present the performance measures summarizing the base metric across $\mathcal{S}$. In the following, we let $\mathcal{B}$ contain all baseline learning algorithms and ICL. We set the test prompt length as $T = 2T_{\mathrm{train}} = 100$, which is within the length generalization regime observed in practice [25].

**Base metric.** Our base metric is the *performance ratio*, which normalizes the sample complexity of a learning algorithm by that of the best algorithm among all baselines.

**Definition 2.1.** For a learning algorithm $b \in \mathcal{B}$ at a scenario $s \in \mathcal{S}$, the *performance ratio* of a requirement $r$ against $\tilde{\mathcal{B}} \subseteq \mathcal{B}$ is defined as $R_b^s(r; \tilde{\mathcal{B}}) = N_b^s(r)/\min_{\tilde{b} \in \tilde{\mathcal{B}}}\{N_{\tilde{b}}^s(r)\}$, where $N_b^s(r) \triangleq \min\{t \mid \mathbb{E}[l(f_b^t(X_{t+1}^s), Y_{t+1}^s)] \leq r\}$ is the sample complexity of achieving the performance $r$.

The performance ratio quantifies how many more demonstrations is required by a learning algorithm to achieve certain performance level compared to the best learner among $\tilde{\mathcal{B}}$. Therefore, when $\mathsf{BMA} \in \tilde{\mathcal{B}}$, algorithms with $R_b^s(r; \tilde{\mathcal{B}}) = 1$ have optimal efficiency at $s$ due to the optimality of $\mathsf{BMA}$.

**Performance measures.** Based on the performance ratio across different scenarios, our goal is to report a "single" score that summarizes how optimal ICL is across $\mathcal{S}$. However, naively summarizing the performance ratio for a requirement $r$ is inappropriate because the difficulty of achieving $r$ varies across learning problems, making comparisons inconsistent. Therefore, we define the *reference performance quantile* $\psi_{\mathcal{B}^{\mathrm{ref}}}^{\mathcal{Q}}(s)$ as the $\mathcal{Q}$-th quantile of reference performances at $s$ for $\mathcal{Q} \in (0, 1)$. Here, we measure the performance quantile in a reverse order, for making higher performance quantile analogous to higher performance. The reference performances at $s$ is defined as a set of performances achieved by reference models $\mathcal{B}^{\mathrm{ref}} \subseteq \mathcal{B}$; that is, $\{\mathbb{E}[l(f_b^t(X_{t+1}^s), Y_{t+1}^s)]|b \in \mathcal{B}^{\mathrm{ref}}, t \in [T]\}$.

With this idea, the performance ratios across $\mathcal{S}$ is summarized by the *mean performance ratio* and the *performance profile*, which are defined as follows.

**Definition 2.2.** For the performance quantile $\psi_{\mathcal{B}^{\mathrm{ref}}}^{\mathcal{Q}}$, the *mean performance ratio* of $b \in \mathcal{B}$ against $\tilde{\mathcal{B}} \subseteq \mathcal{B}$ is defined as $\mathrm{MPR}(b; \psi_{\mathcal{B}^{\mathrm{ref}}}^{\mathcal{Q}}, \tilde{\mathcal{B}}) \triangleq \frac{1}{|\mathcal{S}|}\sum_{s \in \mathcal{S}} R_b^s(\psi_{\mathcal{B}^{\mathrm{ref}}}^{\mathcal{Q}}(s); \tilde{\mathcal{B}})$.

**Definition 2.3.** For the performance quantile $\psi_{\mathcal{B}^{\mathrm{ref}}}^{\mathcal{Q}}$, the *performance profile* of $b \in \mathcal{B}$ against $\tilde{\mathcal{B}} \subseteq \mathcal{B}$ at a ratio $\tau \geq 1$ is defined as

$$ \rho_b(\tau; \psi_{\mathcal{B}^{\mathrm{ref}}}^{\mathcal{Q}}, \tilde{\mathcal{B}}) = \frac{1}{|\mathcal{S}|}|\{s \in \mathcal{S} : R_b^s(\psi_{\mathcal{B}^{\mathrm{ref}}}^{\mathcal{Q}}(s); \tilde{\mathcal{B}}) \leq \tau\}|. $$

The two measures capture complementary aspects of optimality of ICL. Specifically, the mean performance quantile quantifies the average inefficiency of a learning algorithm $b$ in attaining a certain performance, which is assumed to be achievable by $b$. In contrast, the performance profile measures the frequency with which a model $b$ can achieve the performance quantile given a tolerance for inefficiency. These intuitive measures provide novel insights into optimality of ICL that are not apparent in previous error rates-based comparisons and asymptotic analyses.

## 3 Benchmarking ICL Efficiency

We measure to what extents transformers efficiently learn a new task through ICL compared to the optimal learning algorithm (§3.1) and principled baselines (§3.2).

### 3.1 Can Transformer Optimally Learn In Context?

We first examine the efficiency of ICL compared to the Bayes optimal predictor, which learns new concepts with optimal efficiency. For comprehensive evaluation, we design the test scenarios with various levels of SNRs: $\mathcal{S} = \{([M], \sigma_y^2, \sigma_w^2) \mid M = 10, \sigma_y^2 \in \{0.003, 0.03, 0.3\}, \sigma_y^2 \in \{0.1, 1, 10\}\}$ (cf. §2.1). Also, to minimize the impacts of stochasticity of the sampling process of $H_t$, we evaluate performances for each scenario 512 times. Then, we analyze the mean performance ratio of ICL against BMA for all quantiles of performances achieved by ICL; that is, we measure $\mathrm{MPR}(\mathsf{ICL}; \psi_{\mathcal{B}_1^{\mathrm{ref}}}^{\mathcal{Q}}, \tilde{\mathcal{B}}_1)$ with $\mathcal{B}_1^{\mathrm{ref}} \triangleq \{\mathsf{ICL}\}$ and $\tilde{\mathcal{B}}_1 \triangleq \{\mathsf{ICL}, \mathsf{BMA}\}$ for $\mathcal{Q} \in \{0.01, 0.1, \cdots, 0.9, 0.99\}$. In this way, we measure the efficiency of ICL in achieving each performance level under various difficulties in extracting information from prompts. In the following, we regard prompts with more than 40 demonstrations as the many-shot regime where the average performance quantile is approximately 0.5 (cf. Figure A3 in Appendix).

Figure 1 reveals a striking dichotomy in optimality of ICL.

**Near optimal few-show efficiency.** For low performance quantiles ($\mathcal{Q} \leq 0.3$), ICL demonstrates its remarkable near optimal efficiency. Specifically, the mean performance ratio is at most 1.1, which means that it requires only 10% more demonstrations on average than the optimal learning algorithm to achieve the performance lower than $\psi_{\mathcal{B}_1^{\mathrm{ref}}}^{0.3}(s)$ for $s \in \mathcal{S}$. Considering the average sample complexity for the performance quantile of 0.3 is 19, this explains ICL's impressive few-shot performance observed in practice (e.g., demonstration sizes of 5 and 15 in Brown et al. [1]).

**Suboptimal many-shot efficiency.** Starting from $\mathcal{Q} = 0.3$ or more apparently from $\mathcal{Q} = 0.7$ onward, *the performance ratio grows almost monotonically with $\mathcal{Q}$*, increasing from around 1.1 at $\mathcal{Q} = 0.3$ to around 1.2 at $\mathcal{Q} = 0.7$ and to around 1.45 at $\mathcal{Q} = 0.99$. This implies that efficiency of ICL as a learning algorithm deteriorates when pursuing high performance requirements, which inherently requires larger demonstration sizes. Therefore, when scaling ICL from few-shot to many-shot regimes, expecting a similar level of optimality relative to the optimal learner would be an overestimation. Importantly, the diminishing efficiency of ICL would be difficult to uncover without our novel evaluation framework since ICL performances generally improve with more demonstrations [26] and its learning curve resembles that of the Bayes optimal estimator [16].

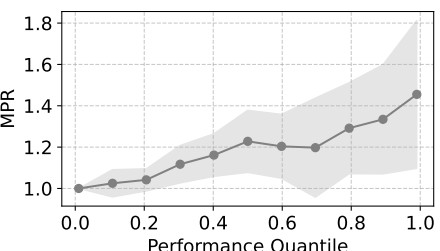

Figure 1: Mean performance ratio of ICL against BMA across different performance requirements. The shaded areas represent the standard deviation of the corresponding performance ratio.

### 3.2 Benchmarking ICL Against Principled Baselines

We have shown that ICL is significantly inefficient compared to BMA in high performance regimes. While BMA is learnable by minimizing (1), it might seem unrealistic for ICL to compete with BMA that performs the expensive model averaging operation. Thus, we compare ICL with more practical baselines with a computational constraint that select a single model using principled criteria (cf. (3)): Akaike Information Criterion (AIC) [27] as a minimax-rate optimal model selection mechanism, Bayesian Information Criterion (BIC) [28] as a consistent model selection mechanism, and Bayesian Model Comparison (BMC) as an efficient BMA alternative selecting maximum a posteriori model class. These baselines represent the spectrum of principled model selection methods, which often asymptotically converge to either AIC or BIC [29].

To quantify relative efficiency, we use performance profiles $\rho_b(\tau; \psi_{\mathcal{B}_2^{\mathrm{ref}}}^{\mathcal{Q}}, \tilde{\mathcal{B}}_2)$ with $\mathcal{B}_2^{\mathrm{ref}} = \{\mathsf{ICL}, \mathsf{AIC}\}$ and $\tilde{\mathcal{B}}_2 = \{\mathsf{ICL}, \mathsf{AIC}, \mathsf{BIC}, \mathsf{BMC}\}$. This allows us to measure the probability that each method achieves a reference performance level within given sample complexity budgets, which evaluates both efficiency and effectiveness (i.e., maximum achievable performances) of learning algorithms.

**Superiority of ICL in few-shot regimes.** Perhaps not surprisingly (given the results from comparison with BMA), ICL dominates the baselines with restricted capacity under low performance requirements.

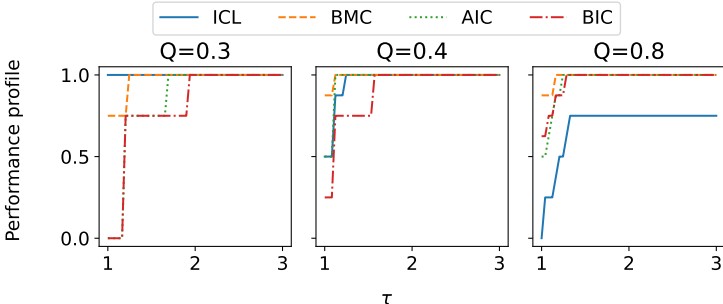

Figure 2: Performance profiles $\rho_b$ across different performance ratios $\tau$ under different target performance quantiles $\mathcal{Q}$. Each curve represents the probability that a method achieves the desired performance within a factor $\tau$ of the best method's sample complexity (x-axes). Figure A4 in Appendix illustrates results for all $\mathcal{Q}$.

Specifically, it achieves the perfect performance profile at $\tau = 1$ for $\mathcal{Q} \leq 0.3$. This means that it optimally attains the performance requirement in *all* scenarios when $\mathcal{Q} \leq 0.3$. Given that each baseline has its own strength in certain scenarios, this guarantee is quite strong and not observed in other baselines. Further, for $\mathcal{Q} = 0.4$, ICL reaches a perfect performance profile within $\tau \leq 1.2$. This means that ICL attains the required performance of $\mathcal{Q} = 0.4$ in all scenarios by using at most 20% more demonstrations on average compared to the best method in each scenario. Conversely, all baselines selecting a single model struggle in the low-performance regime due to high uncertainty under a small number of demonstrations preventing them from selecting the proper model class [30, 31]. The results highlight significance of ICL as a learning algorithm in few-shot regimes.

**Inferiority of ICL in many-shot regimes.** Figure 2 illustrates diminishing efficiency of ICL in long context regimes. Specifically, as the performance requirement increases, the initial performance profile at $\tau = 1$ is reduced, indicating the reduced probability that ICL learns the most efficiently among $\tilde{\mathcal{B}}_2$. Beside, the computational budget $\tau$ required to reach perfect performance profile increases as the performance requirement increases. Eventually for $\mathcal{Q} \geq 0.8$, even at $\tau = 3$, ICL achieves the performance profile around 0.8, which means that ICL cannot reach the performance requirements for 20% of cases by using even 3 times more demonstrations than other models.

Crucially, this increasingly suboptimal behavior is opposite to the behaviors of principled baselines. In Figure 2, as opposed to ICL, the principled learning algorithms significantly reduce the time to reach the (near) perfect performance profiles as $\mathcal{Q}$ increases. Eventually, despite their significant deficiencies in few-shot regimes, all such baselines become more effective (achieving higher performance profiles at $\tau = 3$) and more efficient (sharply improving the performance profiles with respect to $\tau$) than ICL in many-shot regimes. Therefore, some characteristics enabling learning algorithms to leverage large number of demonstrations might be missing in the ICL mechanism.

To gain further insights, we qualitatively analyze MSEs across different numbers of demonstrations for each scenario. As a trivial baseline, we also consider an ensemble that aggregates the ridge estimators $\{\hat{w}_m\}_{m \in [M]}$ using equal weights. Figure 3a shows that while all methods show decreasing MSEs with more demonstrations, ICL exhibits persistent discrepancies from the principled learning algorithms in many-shot regimes. Further, in Figure 3b, we analyze the squared prediction difference between each model and the Bayes optimal predictor for each scenario. Critically, it reveals that while consistent estimators (BMC, BIC) seem to converge in $L^2$ to BMA (albeit at different rates), ICL's $L^2$ distance to BMA plateaus after receiving few demonstrations. This behavior mirrors the trivial ensemble, which does not update its hypothesis about the model class with demonstrations. This suggests another fundamental limitation: ICL may lack asymptotic efficiency and consistency (cf. Ding et al. [29] for formal definitions). These findings challenge the prevailing optimism about ICL's potential as a universal learning algorithm.

### 3.3 On Sources of the Diminishing Efficiency

We observe ICL falls short under high performance requirements, which typically requires longer context sizes than the pretraining prompt (cf. Figure A3 in Appendix). Given universally observed deficiencies of machine learning models in the out-of-distribution (OOD) regimes [32, 33], it is tempting to attribute the diminishing efficiency to the deficiencies in OOD regimes.

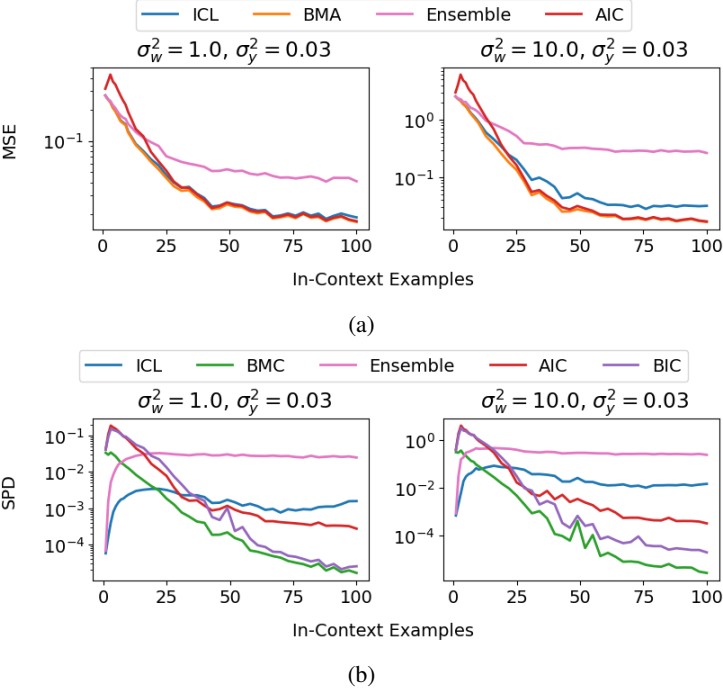

(a)

(b)

Figure 3: **(a)** Mean squared errors for different demonstration sizes. **(b)** Squared prediction differences between BMA and other methods for different demonstration sizes. Figure A5 and A6 in Appendix illustrates results for all $s \in \mathcal{S}$ and $b \in \mathcal{B}$.

We take a closer look at this in Figure 3b, which corresponds to the reducible error due to the bias-variance decomposition. Recalling that $T_{\text{train}} = 50$ was used for pretraining, Figure 3b and Figure A6 in Appendix show no apparent differences in the achievable error between in-distribution and OOD regimes, except in low SNR scenarios $(\sigma_w^2, \sigma_\epsilon^2) = (0.1, 0.03)$ and $(\sigma_w^2, \sigma_\epsilon^2) = (1, 0.3)$. This finding aligns with the length generalization literature, which suggests that transformers often generalize to contexts up to 2.5 times longer than those seen during pretraining [25]. Further, given that the average performance quantile at $T_{\text{train}}$ is 0.6, Figure 1 reveals that fundamental inefficiency already emerges in the in-distribution regime. Therefore, the diminishing efficiency observed in §3.1 and §3.2 cannot be fully attributed to the transformers' OOD generalization capability.

## 4 Analyzing Suboptimality of ICL

In this section, we explain why ICL's efficiency as a learning algorithm diminishes in long context by using information-theoretic tools. This theoretical grounding is crucial, as it shows that the diminishing efficiency of ICL in long context observed in §3 is an inherent property of the ICL mechanism itself, rather than an artifact of a specific experimental setup. Accurately identifying the source of this limitation is essential for guiding future efforts to mitigate it.

### 4.1 ICL Error Decomposition

Adopting a Bayesian viewpoint [9], we denote the oracle distribution with $e$ drawn from an environment $\mathcal{E}$ by $\bar{P}_e^t(\cdot) \triangleq \mathbb{P}(Y_{t+1} \in \cdot | H_t, e) = \mathbb{P}(Y_{t+1} \in \cdot | X_{t+1}, e)$ (e.g., $\mathcal{E}$ characterizes the sampling process in §2.1 with $e = (m, w_m)$). Similarly, we let $\text{TF}_\theta$ models the conditional distribution of outputs, i.e., $\text{TF}_\theta(H_t) \triangleq P_\theta(Y_{t+1} \in \cdot \mid H_t) \triangleq P_\theta^t(\cdot)$. All subsequent discussions in this section assumes no distribution shift; that is, $\mathcal{E}$ is the environment under which $\text{TF}_\theta$ was pretrained. We assume that $Y_{t+1}$ is either discrete or continuous.

With this notation, the ICL performance with $t$ demonstrations from $\mathcal{E}$ is defined as $\mathbb{E}\left[-\log P_\theta^t(Y_{t+1})\right] = \mathbb{E}\left[-\log \bar{P}_e^t(Y_{t+1})\right] + \mathbb{E}\left[D_{\text{KL}}(\bar{P}_e^t \parallel P_\theta^t)\right]$ [9]. Here, the first term is the (irreducible) aleatoric uncertainty and constant with respect to $t$ in our setting. The second term can

be further decomposed as

$$\mathbb{E}\left[D_{\mathrm{KL}}(\bar{P}_e^t \parallel P_\theta^t)\right] = \mathbb{E}\left[\int \log \frac{d\bar{P}_e^t}{dP_\theta^t}(y)\bar{P}_e^t(dy)\right] = \underbrace{\mathbb{E}\left[D_{\mathrm{KL}}(\bar{P}_e^t \parallel \hat{P}_\mathcal{E}^t)\right]}_{\triangleq \epsilon_{\mathrm{Bayes}}^t(\text{Bayes risk})} + \underbrace{\mathbb{E}\left[\log \frac{\hat{P}_\mathcal{E}^t(Y_{t+1})}{P_\theta^t(Y_{t+1})}\right]}_{\triangleq \epsilon_{\mathrm{XS}}^t(\text{Excess risk})}, \quad (4)$$

where the second equality comes from the law of total expectation and $\hat{P}_\mathcal{E}^t(Y_{t+1}) \triangleq \mathbb{P}(Y_{t+1} \in \cdot | H_t, \mathcal{E})$ is the posterior over $Y_{t+1}$ given $H_t$.

In (4), the Bayes risk $\epsilon_{\mathrm{Bayes}}^t$ measures how well the Bayes-optimal predictor performs under uncertainty on $e$. It is non-negative and *decreases monotonically* with more demonstrations; that is, $\epsilon_{\mathrm{Bayes}}^{t+1} \leq \epsilon_{\mathrm{Bayes}}^t$ for all $t \in \mathbb{N}$ [34]. Demonstration size $t$ required to bring this risk below a threshold $q$ is captured by $N_{\mathrm{BMA}}(q) \triangleq \min_{t \in \mathbb{N}}\{\epsilon_{\mathrm{Bayes}}^t \leq q\}$. Here, $q$ represents the absolute value of the performance requirement (e.g., MSE), whereas $\mathcal{Q}$ in §3 denotes the performance quantile.

The excess risk $\epsilon_{\mathrm{XS}}^t$ measures the performance of the transformer relative to the Bayes optimal predictor. Due to the non-negativity of excess risk and independence between $\mathrm{TF}_\theta$ and $\epsilon_{\mathrm{Bayes}}^t$, this term determines when ICL emerges and how well it can perform. For instance, if $\mathrm{TF}_\theta$ achieves an excess risk curve such that $\epsilon_{\mathrm{XS}}^t - \epsilon_{\mathrm{XS}}^0 \leq \epsilon_{\mathrm{Bayes}}^0 - \epsilon_{\mathrm{Bayes}}^t$, non-trivial ICL performance emerges, improving upon the zero-shot performance with demonstrations. Further, if $\epsilon_{\mathrm{XS}}^t \to 0$ as $t \to \infty$, then ICL is Bayes-risk consistent and asymptotically matches BMA. In §4.2, we dissect the excess risk $\epsilon_{\mathrm{XS}}^t$ based on empirical observations in §3.

## 4.2 On Excess Risk

Interpreting the transformer's prediction in the meta ICL setup as the Gaussian distribution (e.g., by adding a small random Gaussian noise to the prediction), the squared prediction difference in Figure 3b is directly proportional to the excess risk, up to a constant scale and shift. The same applies to each baseline's squared prediction difference, interpreted as its own excess risks.

In this regard, Figure 3b illustrates that the transformer's excess risk remains roughly bounded within a modest interval in a certain length generalization regime (e.g., $t \leq 2T_{\mathrm{train}}$), suggesting that it would perform ICL non-trivially due to the monotonicity of $\epsilon_{\mathrm{Bayes}}^t$. However, once the context length becomes much longer than the one seen during pretraining (e.g., $t > 2T_{\mathrm{train}}$ in Figure 4), the excess risk deteriorates sharply. This explains why ICL is not a consistent learner, being dominated by the principled learning algorithms in large demonstration regimes, as we observed in §3.2. We formally encode the above empirical observations about the non-vanishing excess risk curve as follows.

**Assumption 4.1.** For an environment $\mathcal{E}$ and a transformer $TF_\theta$, there exist constants $(\bar{t}, \triangle_{\mathrm{XS}}) \in (\mathbb{N}, \mathbb{R}_+)$ such that $0 \leq \triangle_{\mathrm{XS}} \leq \epsilon_{\mathrm{XS}}^{t'}$ for all $t' \geq \bar{t}$.

The assumption states that, after some reference point $\bar{t}$, the excess risks of $\mathrm{TF}_\theta$ can be lower bounded, aligning with the behaviors illustrated in Figures 3b and 4 as well as with empirical evidence demonstrating the deficiencies of state-of-the-art LLMs outside the length generalization regime [35, 25]. In other words, it assumes that $\mathrm{TF}_\theta$ does not magically reduce its excess risk in the OOD context length regimes. This may occur for various reasons such as insufficient pretraining data or intrinsic properties of architectures. Importantly, we do not assume conditions on the cause, only that the lower bound exists. We also emphasize that, while the excess risk is lower bounded under Assumption 4.1, ICL performance can still improve with more demonstrations due to the monotonicity of the Bayes risk (cf. §4.1) as observed in many-shot ICL literature [36, 26].

Crucially, as we show in §4.3, $\triangle_{\mathrm{XS}}$ controls a lower bound of ICL's suboptimal efficiency in learning from demonstrations. For a transformer with a strong length generalization ability, $\epsilon_{\mathrm{XS}}^{t'}$ in the assumption can also be upper bounded, making the subsequent suboptimality analysis nearly tight. In this regard, our analysis encompasses plausible (near) future advances in length generalization capability. Therefore, our analysis under Assumption 4.1 is a general result highlighting the ICL mechanism's intrinsic flaws, isolating them from the transformer's length generalization capability.

## 4.3 Analyzing Suboptimality of ICL

Next, we explain the critical suboptimality of ICL observed in §3, where ICL initially matches the efficiency of the optimal learning algorithm but starts to significantly deteriorate in many-shot regimes.

To this end, we define suboptimality of ICL at performance requirement $q$ as the additional number of demonstrations required for ICL to achieve requirement $q$ compared to the Bayes optimal estimator, denoted as $\texttt{SubOpt}(q) \triangleq \min_t \{t - N_{\mathsf{BMA}}(q) \mid \epsilon_{\mathrm{Bayes}}^t + \epsilon_{\mathrm{XS}}^t \leq q\}$. Here, we define suboptimality at $q$ with respect to the reducible part of the ICL performance (i.e., $\mathbb{E}\left[D_{\mathrm{KL}}(\bar{P}_e^t \parallel P_\theta^t)\right]$), which is equivalent to defining it with respect to the ICL performance up to constant scaling in $q$.

The following theorem constructs a lower bound of $\texttt{SubOpt}(q)$ under Assumption 4.1 where $\mathbb{I}$ denotes the mutual information.

**Theorem 4.2.** *Let us assume $(\bar{t}, \triangle_{XS})$ satisfies Assumption 4.1. For a sufficiently small $q$ such that $N_{\mathsf{BMA}}(q) \geq \bar{t}$, it holds that*

$$\texttt{SubOpt}(q) \geq LB(q) \triangleq \min_{t \in \mathbb{N}} \left\{ t \mid \mathbb{I}(Y_{N_{\mathsf{BMA}}(q)}; \tilde{D}_{t+1} \mid H_{N_{\mathsf{BMA}}(q)-1}) > \triangle_{XS} \right\} \tag{5}$$

*where $\tilde{D}_{t+1}$ is a sample from the same distribution as $D_{t+1}$.*

Theorem 4.2 intuitively characterizes suboptimality (cf. Figure A2 in Appendix for an illustration of the concept). Specifically, suppose the Bayes optimal learner requires $N_{\mathsf{BMA}}(q)$ demonstrations to achieve the performance $q$. Then, $\texttt{SubOpt}(q)$ represents the additional demonstrations required for ICL to compensate for the excess risk $\epsilon_{\mathrm{XS}}^t$. Here, the compensation represents how much the new demonstrations $\tilde{D}_{t+1}$ reduce the uncertainty about $Y_{N_{\mathsf{BMA}}(q)}$ given a prompt $H_{N_{\mathsf{BMA}}(q)-1}$, which corresponds to the conditional mutual information in (5). The theorem is proven in §B.1.

Characterizing suboptimality with $\mathbb{I}(Y_{N_{\mathsf{BMA}}(q)}; \tilde{D}_{t+1} \mid H_{N_{\mathsf{BMA}}(q)-1})$ provides clear insights into ICL's suboptimality. Specifically, transformers with small excess risks in the non-vanishing regime are less subject to suboptimality. Besides, since a higher performance requirement (i.e., a smaller $q$) increases $N_{\mathsf{BMA}}(q)$, suboptimality naturally increases due to reduced conditional mutual information. The following theorem, which is proven in §B.2, makes this intuition precise by establishing necessary conditions for $\texttt{SubOpt}(q)$ being constant with respect to $q$.

**Theorem 4.3.** *Let us assume $(\bar{t}, \triangle_{XS})$ satisfies Assumption 4.1 and let $q$ be such that $N_{\mathsf{BMA}}(q) \geq \bar{t}$. If $LB(q') = LB(q)$ for all $\triangle_{XS} < q' < q$, then either of the following condition holds:*

1. ***Negligible excess risk:*** $\triangle_{XS} \leq \mathbb{I}(Y_t; \tilde{D}_1 | H_{t-1})$ *for all $t \geq N_{\mathsf{BMA}}(q)$, and $LB(q) = 0$,*

2. ***Negligible diminishing returns:*** $\mathbb{I}(Y_{\tilde{t}}; \tilde{D}_1 | H_{\tilde{t}-1}) < \left(1 + \frac{1}{LB(q)}\right) \mathbb{I}(Y_t; \tilde{D}_1 | H_{t-1})$ *for all $t \geq N_{\mathsf{BMA}}(q)$, where $\tilde{t} \triangleq N_{\mathsf{BMA}}(q) + LB(q)$ and $LB(q) > 0$.*

Non-deteriorating suboptimality has stringent necessary conditions that rarely hold in practice. Specifically, the *negligible excess risk* condition requires that the information gain from a single demonstration, regardless of demonstration size, dominates the excess risk. While this may hold for few-shot regimes (explaining the significant efficiency of few-shot ICL), ensuring this assumption across all prompt lengths is quite strong given the diminishing nature of $\mathbb{I}(Y_t; \tilde{D}_1 \mid H_{t-1})$ with $t$ in most learning scenarios [37, 38]. For a similar reason, the *negligible diminishing returns* condition, which requires a constant lower bound of $\mathbb{I}(Y_t; \tilde{D}_1 \mid H_{t-1})$ for all demonstration sizes $t$, is quite strong. Therefore, $\texttt{SubOpt}(q)$ inevitably grows as $q$ decreases, leading to increasing suboptimality of ICL under a high performance requirement as observed in §3.

As a concrete intuition on suboptimality, we consider the following crude approximations: (A1) $\epsilon_{\mathrm{Bayes}}^t \approx C_1/\sqrt{t}$ for some constant factor $C_1$ and (A2) $\epsilon_{\mathrm{XS}} \lesssim \epsilon_{\mathrm{XS}}^t$ for all $t \in \mathbb{N}_+$. Here, (A1) corresponds to sublinear convergence of the Bayes posterior estimator, which holds in many cases [37, 38], and (A2) corresponds to Assumption 4.1 with $(\bar{t}, \triangle_{XS}) = (0, \epsilon_{\mathrm{XS}})$. Replacing (A1) with other common bounds, such as $\epsilon_{\mathrm{Bayes}}^t \approx C_1/t$ or $\epsilon_{\mathrm{Bayes}}^t \approx C_1 \exp(-t)$, yields similar results.

Under (A1) and (A2), for performance achievable by the transformer (i.e., $q > \epsilon_{\mathrm{XS}}$), a simple calculation gives $\texttt{SubOpt}(q) \gtrsim \frac{C_1^2}{(q-\epsilon_{\mathrm{XS}})^2} - \frac{C_1^2}{q^2} \geq \frac{C_1^2 \epsilon_{\mathrm{XS}}}{q^2(q-\epsilon_{\mathrm{XS}})}$. Here, the rapid growth of $\texttt{SubOpt}(q)$ as $q$ decreases highlights the inefficiency of ICL in achieving high performance requirement. Moreover, another way of improving suboptimality by reducing $\epsilon_{\mathrm{XS}}$, from the perspective of the rough power law estimations from the scaling laws [39], would require an exponential increase in pretraining data size or computational resources. Thus, in either way, a transformer exhibits significant suboptimality in achieving high performance through ICL compared to principled learning algorithms.

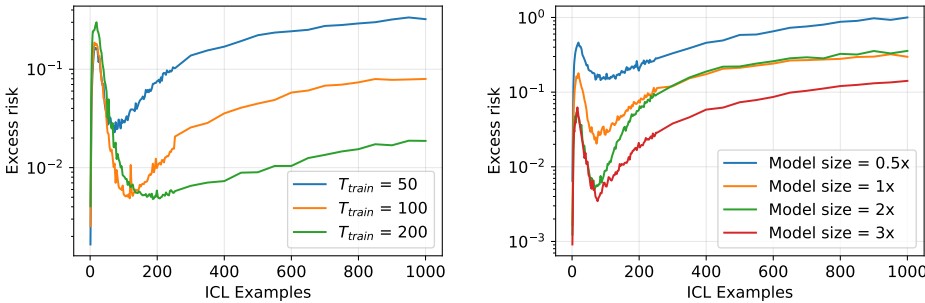

Figure 4: Impacts of the pretraining prompt length (left) and the model size (right) on the excess risk curve in $\mathcal{E} = ([M], \sigma_y^2, \sigma_w^2) = ([10], 0.03, 10)$.

## 4.4 Impacts of Scaling Computations

Given that the non-vanishing excess risk curve causes an inherent inefficiency of ICL in long context, we explore whether improving transformers' capacity of handling longer context can enable the excess risk to decrease with more demonstrations, thus overcoming this fundamental limitation. As illustrated in Figure 4, while larger models and longer pretraining prompt lengths reduce the magnitude of the excess risk in many-shot regimes, the non-vanishing shape in long contexts persists. Thus, simply scaling model size or pretraining prompt length does not fundamentally resolve the inefficiency of ICL in long contexts. See Appendix A.3 for experimental settings and detailed results.

## 5 Related Work

**Asymptotic Behavior Analysis.** Xie et al. [2] show that ICL predictions converge to posterior probabilities in asymptotic demonstration size regimes. Subsequent works expand these results to encompass finite-sample guarantees [12, 10, 11], broader prompt distribution structures [12, 40, 10], and structural characteristics of transformers [10]. Recent studies analyze the average cumulative regret across demonstrations [10, 9], treating ICL as an online learning algorithm. However, practical applications prioritize test sample performance over demonstration set performance. In this work, we directly analyze suboptimality of ICL in achieving a specific performance requirement through the excess sample complexity compared to the Bayes optimal learning algorithm.

**Stylized ICL Benchmarks.** With the meta ICL framework (cf. §2.1), Garg et al. [16] demonstrate that transformers are capable of learning simple function classes (e.g., linear models and random neural networks) from demonstrations, achieving error curves qualitatively similar to those of optimal learning algorithms under asymptotic pretraining sample conditions. Subsequent works extend the results to finite pretraining sample scenarios [6] and mixture function classes [41, 21]. Further, new analytical frameworks that directly analyze ICL predictions reveal that ICL exhibits behavior similar to gradient descent [19, 18]. More recently, these stylized settings have been used to probe other sophisticated behaviors of ICL. This includes analyzing transformers' in-context model selection and preference for simpler hypotheses [42, 43], their ability to infer causal structures [44], and the implicit connection between ICL and low-rank updates to MLP layers [45]. Although stylized ICL benchmarks have been extensively studied, the optimality of ICL as a learning algorithm remains unexplored. By comparing the sample complexity of ICL with that of principled learning algorithms, we uncover a novel insight on the fundamental inefficiency of ICL in the many-shot learning regime. This critical insight suggests a more nuanced view of ICL as a purported universal problem solver.

## 6 Conclusion

The surprisingly strong ICL performance of LLMs suggest its potential to eliminate the need for task-specific models. To rigorously examine this potential, we developed a novel framework for benchmarking optimality of ICL as a learning algorithm against principled learning algorithms. We found that while few-shot ICL's efficiency is comparable to the Bayes optimal learning algorithm, its efficiency quickly diminishes with more demonstrations. Through information-theoretic analyses, we showed that ICL mechanism is intrinsically inefficient in many-shot regimes. This highlights the need for a new adaptation method that can reduce excess risk with more demonstrations, enabling sample-efficient learning of novel tasks while preserving the update-free nature of ICL.

## Acknowledgments and Disclosure of Funding

We would like to thank Jihyeon Hyeong, Yuchen Lou, and anonymous reviewers for their valuable feedback during the preparation of this manuscript. We declare that there was no funding received for this work.

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

# A    Additional Details

## A.1    On Usage of Stylized Setting

**Comprehensive analyses with statistical significance.** The benchmark in the stylized settings in principle enables comprehensive comparisons across different environments (e.g., $\mathcal{S}$) and architectures (e.g., different $\text{TF}_\theta$), achieving arbitrarily high levels of statistical significance. In empirical studies, these factors are constrained to the configurations of the datasets or the computational budgets.

**Comparison with the optimal method.** The stylized setting enables comparison with principled learning algorithms. Specifically, BMA considered in (2) provides the minimum achievable performance of any learning algorithms *at all prompt lengths*. This strong guarantee is typically not possible in empirical studies, as even human performances could not be an oracle or simply not possible to attain with only the data provided to the transformer. Also, the theoretical studies themselves do not allow for precise performance comparison, except analyzing the general asymptotic behavior that is shared among reasonable learning algorithms.

**From stylized settings to practical LLMs.** Although we study stylized settings in a rigorous manner, it does not capture all aspects of LLMs. For example, the ICL objective in (1) is not an autoregressive loss used for pretraining LLMs, omitting the losses of predictions at each $Y_t$. Further, the model size and training data diversity used in our meta ICL setup are significantly smaller and less diverse than those used in modern LLMs, such as GPT-4 [46] and Gemini 2.5 [47], which typically possess hundreds of billions or even trillions of parameters trained on vast, heterogeneous datasets. Therefore, one potential concern is the generalization of results obtained in stylized settings. While it cannot be shown precisely, the findings from such stylized settings have been surprisingly well generalized to the real-world tasks [48, 40]. For instance, Ahn et al. [48] perform synthetic experiments even with simplified transformers to study optimization methods for LLMs that surprisingly well reproduce the results from the real-world natural language data.

Given the significance of actionable insights from the stylized settings such as foretelling impacts of scaling ICL to the asymptotic region of the demonstration size, which is extremely challenging with real-world LLMs, we hold positive views on the role of stylized settings in LLM research whose significant advantages outweigh the potential concerns on its generalization to the LLMs in practice.

## A.2    Detailed Configurations

**Model.**    For the model, we use the GPT-2 [22] architecture for $\text{TF}_\theta$, which is a standard architecture in the meta ICL and other stylized experimental settings; that is, we define $\text{TF}_\theta$ as a decoder-only transformers [49] with 12 layers, 8 attention heads, and 256-dimensional embedding space. For readers unfamiliar with transformers, we refer to the excellent tutorials [50, 51]. We remark that viewing $\text{TF}_\theta$ as a function from a sequence of vectors with an arbitrary length to a vector with the same dimension does not significantly impact the understanding of core findings in this paper.

**Optimization.**    For minimizing the ICL objective $l(\theta)$, we compute the stochastic gradient with 64 prompts and update $\theta$ by using the Adam optimizer [52] with fixed learning rate of $10^{-4}$ for one million training iterations. Also, in order to boost the convergence speed, we use curriculum learning [53] as recommended in [16, 21] by increasing the length of the prompt by 2 every 2,000 training iterations until it reaches $(2M + 1)$ (and the order of Fourier series by 1 until it reaches $M$).

**Computational resources for experiments.**    In this work, we use multiple servers which consist of multiple GPUs including RTX 8000 (50GB) and A100 (40GB).

## A.3    Impacts of Scaling Computations

We show that the non-vanishing excess risk curve of the transformer in long context causes the efficiency of learning to diminish with more demonstrations. Therefore, a natural question is whether enhancing transformers' capacity of handling longer context can make the excess risk decrease with more demonstrations and thus resolve the fundamental inefficiency. We analyze the impacts of scaling the pretraining context lengths (by setting $T_{\text{train}}$ to 100 and 200) and the model sizes (by scaling the number of layers, the number of heads, the embedding dimension by factors of 0.5, 2, and 3) on the excess risk. The pretraining losses are 1.06, 0.58, and 0.34 for models trained with $T_{\text{train}} = 50$, $T_{\text{train}} = 100$, and $T_{\text{train}} = 200$, respectively. Also, the pretraining losses are 1.36, 1.19, 0.99, and 0.90

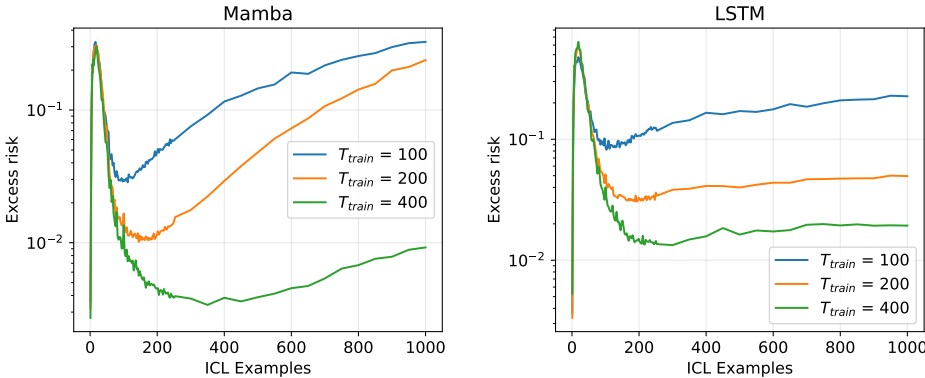

Figure A1: The excess risk curve in $\mathcal{E} = ([M], \sigma_y^2, \sigma_w^2) = ([10], 0.03, 10)$ under different pretraining prompt length for Mamba and LSTM.

for half-capacity, standard, double-capacity, and triple-capacity models respectively. Note that we did not explore different positional encoding methods since we already use no positional encoding scheme that is effective at length generalization [35, 54], which is from an inductive bias for the sample order-stable learning algorithms.

Figure 4 (left) shows that increasing $T_{\text{train}}$ significantly reduces the excess risk values, especially for long-context regimes as desired. However, overall shape of the excess risk curve remains non-vanishing in the long-context regime. We observe from Figure 4 (right) similar effects of increasing the model sizes. Interestingly, larger models do not increase the length generalization regime, which is consistent with previous results [25].

The results suggest that simply increasing computations with a larger model and a longer pretraining prompt length does not fundamentally change the *shape* of the excess risk, even though their overall scales improve. Therefore, while the degree of suboptimality can be relaxed with reduced excess risk, the inefficiency in many-shot regimes persist.

### A.4 Impacts of Different Architectures

We examine whether the non-vanishing excess risk curve in long context is unique to transformer architecture. To this end, we ran additional experiments with Mamba [55] and LSTM [56] architectures under the setup same as in §A.3. Figure A1 shows that the shapes of the excess risk curve under Mamba and LSTM remain non-vanishing in the long-context regime, being consistent with recent work analyzing the length extrapolation limits of Mamba [57]. Crucially, this means that the the diminishing efficiency of ICL in long context is a general property of the ICL paradigm in sequence models having a non-vanishing excess risk, not an artifact of the transformer architecture.

## B Proof of Claims

### B.1 Proof of Theorem 4.2

*Proof.* We first characterize suboptimality by the Bayes risk as follows:

$$\texttt{SubOpt}(q) = \min_{t \in \mathbb{Z}_+} \left\{ t - N_{\text{BMA}}(q) \mid \epsilon_{\text{Bayes}}^t + \epsilon_{\text{XS}}^t \leq q \right\} \tag{6}$$

$$= \min_{t \in \mathbb{Z}_+} \left\{ t \mid \epsilon_{\text{Bayes}}^t \leq q - \epsilon_{\text{XS}}^t \right\} - N_{\text{BMA}}(q). \tag{7}$$

Since $q < \epsilon_{\text{Bayes}}^{N_{\text{BMA}}(q)-1}$, the monotonicity of $\epsilon_{\text{Bayes}}^t$ and the non-negativity of $\epsilon_{\text{XS}}^t$ give

$$\min_{t \in \mathbb{Z}_+} \left\{ t \mid \epsilon_{\text{Bayes}}^t \leq q - \epsilon_{\text{XS}}^t \right\} = \min_{t \geq N_{\text{BMA}}(q)} \left\{ t \mid \epsilon_{\text{Bayes}}^t \leq q - \epsilon_{\text{XS}}^t \right\} \geq \min_{t \geq N_{\text{BMA}}(q)} \left\{ t \mid \epsilon_{\text{Bayes}}^t < \epsilon_{\text{Bayes}}^{N_{\text{BMA}}(q)-1} - \epsilon_{\text{XS}}^t \right\}.$$
$$\tag{8}$$

To prove the theorem, we note the following.

**(N1). Bayes error reduction as the conditional mutual information**: The Bayes error can be expressed as the reduction of (differential) entropy as follows.

$$\epsilon_{\text{Bayes}}^t = \mathbb{E}\left[ D_{\text{KL}}(\bar{P}_e^t \parallel \hat{P}_{\mathcal{E}}^t) \right] = h(Y_{t+1}|H_t) - h(Y_{t+1}|H_t, e), \ \text{ for continuous } Y_{t+1} \quad (9)$$

$$\epsilon_{\text{Bayes}}^t = \mathbb{E}\left[ D_{\text{KL}}(\bar{P}_e^t \parallel \hat{P}_{\mathcal{E}}^t) \right] = \mathbb{H}(Y_{t+1}|H_t) - \mathbb{H}(Y_{t+1}|H_t, e), \ \text{ for discrete } Y_{t+1} \quad (10)$$

where $h$ is the differential entropy and $\mathbb{H}$ is the Shannon entropy.

Therefore, for any $u \leq v$ and continuous $Y_{t+1}$, we have

$$\epsilon_{\text{Bayes}}^u - \epsilon_{\text{Bayes}}^v = h(Y_{u+1}|H_u) - h(Y_{u+1}|H_u, e) - (h(Y_{v+1}|H_v) - h(Y_{v+1}|H_v, e)) \quad (11)$$

$$= h(Y_{u+1}|X_{u+1}, D_u) - h(Y_{v+1}|X_{v+1}, D_v) \quad (12)$$

$$= \mathbb{I}(Y_{u+1}; \tilde{D}_{v-u}|X_{u+1}, D_u) \quad (13)$$

where $\tilde{D}_{v-u} \triangleq (\tilde{X}_1, \tilde{Y}_1, \cdots, \tilde{X}_{v-u}, \tilde{Y}_{v-u})$ is independently sampled from the same distribution as $D_{v-u}$, the second equality comes from the conditional independence $Y_{n+1} \perp D_n|X_{n+1}, e$ for any $n \in \mathbb{N}_+$, and the last equality comes from the chain rule. For the discrete $Y$'s, the same process can be applied by replacing $h$ with $\mathbb{H}$.

**(N2). Lower bound of the excess risk**: Let $q$ be such that $N_{\text{BMA}}(q) \geq \bar{t}$. Therefore, by Assumption 4.1, we have

$$\left\{ t \in \mathbb{N} \mid t \geq N_{\text{BMA}}(q), \epsilon_{\text{Bayes}}^t < \epsilon_{\text{Bayes}}^{N_{\text{BMA}}(q)-1} - \epsilon_{\text{XS}}^t \right\} \subseteq \left\{ t \in \mathbb{N} \mid t \geq N_{\text{BMA}}(q), \epsilon_{\text{Bayes}}^{N_{\text{BMA}}(q)-1} - \epsilon_{\text{Bayes}}^t > \triangle_{\text{XS}} \right\}.$$
$$(14)$$

By applying **(N1)** and **(N2)** to (7), we get the desired result as

$$\texttt{SubOpt}(q) = \min_{t \geq N_{\text{BMA}}(q)} \left\{ t \mid \epsilon_{\text{Bayes}}^t \leq q - \epsilon_{\text{XS}}^t \right\} - N_{\text{BMA}}(q)$$

$$\geq \min_{t \geq N_{\text{BMA}}(q)} \left\{ t \mid \epsilon_{\text{Bayes}}^{N_{\text{BMA}}(q)-1} - \epsilon_{\text{Bayes}}^t > \triangle_{\text{XS}} \right\} - N_{\text{BMA}}(q) = \min_{t \in \mathbb{N}} \left\{ t \mid \epsilon_{\text{Bayes}}^{N_{\text{BMA}}(q)-1} - \epsilon_{\text{Bayes}}^{t+N_{\text{BMA}}(q)} > \triangle_{\text{XS}} \right\}$$

$$= \min_{t \in \mathbb{N}} \left\{ t \mid \mathbb{I}(Y_{N_{\text{BMA}}(q)}; \tilde{D}_{t+1} \mid H_{N_{\text{BMA}}(q)-1}) > \triangle_{\text{XS}} \right\}. \quad (15)$$

$\square$

## B.2 Proof of Theorem 4.3

*Proof.* Consider $q_1, q_2 \in (\triangle_{\text{XS}}, q)$ such that $q_1 < q_2 < q$ and $N_{\text{BMA}}(q_1) > N_{\text{BMA}}(q_2)$. The goal is to show necessary conditions for $LB(q_1) \leq LB(q_2)$.

Note that $LB(q_1) < LB(q_2)$ is impossible because $\mathbb{I}(Y_{N_{\text{BMA}}(q_1)}; \tilde{D}_{t+1}|H_{N_{\text{BMA}}(q_1)-1}) \leq \mathbb{I}(Y_{N_{\text{BMA}}(q_2)}; \tilde{D}_{t+1}|H_{N_{\text{BMA}}(q_2)-1})$ for any $t \in \mathbb{N}$. Specifically, we have

$$\mathbb{I}(Y_{N_{\text{BMA}}(q_1)}; \tilde{D}_{t+1}|H_{N_{\text{BMA}}(q_1)-1}) \leq \mathbb{I}(Y_{N_{\text{BMA}}(q_2)}; \tilde{D}_{t+1}|H_{N_{\text{BMA}}(q_2)-1}), \quad \forall t \in \mathbb{N} \quad (16)$$

, which implies

$$\left\{ t \in \mathbb{N} \mid \mathbb{I}(Y_{N_{\text{BMA}}(q_1)}; \tilde{D}_{t+1}|H_{N_{\text{BMA}}(q_1)-1}) > \triangle_{\text{XS}} \right\} \subseteq \left\{ t \in \mathbb{N} \mid \mathbb{I}(Y_{N_{\text{BMA}}(q_2)}; \tilde{D}_{t+1}|H_{N_{\text{BMA}}(q_2)-1}) > \triangle_{\text{XS}} \right\}$$
$$(17)$$

, and in turn $LB(q_1) \geq LB(q_2)$.

Therefore, we next show the necessary condition for $LB(q_1) = LB(q_2)$.

**(NC 1). Negligible excess risk**: Let us suppose $\triangle_{\text{XS}} \leq \mathbb{I}(Y_{N_{\text{BMA}}(q_1)}; \tilde{D}_1|H_{N_{\text{BMA}}(q_1)-1}) \leq \mathbb{I}(Y_{N_{\text{BMA}}(q_2)}; \tilde{D}_1|H_{N_{\text{BMA}}(q_2)-1})$. In this case, $LB(q_1) = LB(q_2) = 0$ as desired. Since $q_1$ and $q_2$ are chosen arbitrary, the first necessary condition is given by

$$\triangle_{\text{XS}} \leq \mathbb{I}(Y_t; \tilde{D}_1|H_{t-1}), \quad t \geq \bar{t}. \quad (18)$$

**(NC 2). No diminishing returns**: If **(NC 1)** does not hold, we have $\triangle_{\text{XS}} > \mathbb{I}(Y_{N_{\text{BMA}}(q_1)}; \tilde{D}_1|H_{N_{\text{BMA}}(q_1)-1})$. In this case, we rule out the possibility

$\mathbb{I}(Y_{N_{\text{BMA}}(q_1)}; \tilde{D}_1 | H_{N_{\text{BMA}}(q_1)-1}) \;<\; \triangle_{\text{XS}} \;\leq\; \mathbb{I}(Y_{N_{\text{BMA}}(q_2)}; \tilde{D}_1 | H_{N_{\text{BMA}}(q_2)-1})$ because this gives $LB(q_2) = 0$ and $LB(q_1) > 0$, which contradicts $LB(q_1) = LB(q_2)$.

Thus, we consider the case $\mathbb{I}(Y_{N_{\text{BMA}}(q_1)}; \tilde{D}_1 | H_{N_{\text{BMA}}(q_1)-1}) \leq \mathbb{I}(Y_{N_{\text{BMA}}(q_2)}; \tilde{D}_1 | H_{N_{\text{BMA}}(q_2)-1}) < \triangle_{\text{XS}}$. In this case, $LB(q_1) = LB(q_2)$ requires the following condition

$$\mathbb{I}(Y_{N_{\text{BMA}}(q_2)}; \tilde{D}_{LB(q_2)} | H_{N_{\text{BMA}}(q_2)-1}) < \mathbb{I}(Y_{N_{\text{BMA}}(q_1)}; \tilde{D}_{LB(q_2)+1} | H_{N_{\text{BMA}}(q_1)-1}), \tag{19}$$

where the condition comes from $\mathbb{I}(Y_{N_{\text{BMA}}(q_1)}; \tilde{D}_{t+1} | H_{N_{\text{BMA}}(q_1)-1}) \leq \mathbb{I}(Y_{N_{\text{BMA}}(q_2)}; \tilde{D}_{t+1} | H_{N_{\text{BMA}}(q_2)-1})$ for any $t \in \mathbb{N}$.

By the construction of $q_1$ and $q_2$, we get

$$\mathbb{I}(Y_{N_{\text{BMA}}(q)}; \tilde{D}_{LB(q)} | H_{N_{\text{BMA}}(q)-1}) < \mathbb{I}(Y_{N_{\text{BMA}}(q)+k}; \tilde{D}_{LB(q)+1} | H_{N_{\text{BMA}}(q)-1+k}), \quad \forall k \in \mathbb{N}_+. \tag{20}$$

Due to the chain rule of the mutual information, for any $\tilde{k} \in \mathbb{N}_+$, it holds that

$$\mathbb{I}(Y_{N_{\text{BMA}}(q)}; \tilde{D}_{\tilde{k}} | H_{N_{\text{BMA}}(q)-1}) = \sum_{i=0}^{\tilde{k}-1} \mathbb{I}(Y_{N_{\text{BMA}}(q)+i}; \tilde{D}_1 | H_{N_{\text{BMA}}(q)-1+i}) \geq \tilde{k}\,\mathbb{I}(Y_{N_{\text{BMA}}(q)+\tilde{k}-1}; \tilde{D}_1 | H_{N_{\text{BMA}}(q)+\tilde{k}-2}). \tag{21}$$

Similarly,

$$\mathbb{I}(Y_{N_{\text{BMA}}(q)+k}; \tilde{D}_{\tilde{k}+1} | H_{N_{\text{BMA}}(q)-1+k}) = \sum_{i=0}^{\tilde{k}} \mathbb{I}(Y_{N_{\text{BMA}}(q)+k+i}; \tilde{D}_1 | H_{N_{\text{BMA}}(q)-1+k+i})$$
$$\leq (1+\tilde{k})\mathbb{I}(Y_{N_{\text{BMA}}(q)+k}; \tilde{D}_1 | H_{N_{\text{BMA}}(q)-1+k}). \tag{22}$$

Therefore, we get the second necessary condition as

$$\mathbb{I}(Y_t; \tilde{D}_1 | H_{t-1}) \leq \mathbb{I}(Y_{\bar{t}+\tilde{k}-1}, \tilde{D}_1 | H_{\bar{t}+\tilde{k}-2}) < \left(1 + \frac{1}{\tilde{k}}\right) \mathbb{I}(Y_t; \tilde{D}_1 | H_{t-1}), \quad \forall t \geq \bar{t}, \tag{23}$$

where $\tilde{k} = LB(q) > 1$ for $q$ such that $N_{\text{BMA}}(q) \geq \bar{t}$.

$\square$

## C   Additional Figures

Figure A2: Graphical illustration of Theorem 4.2 when $q = 0.08 - \sigma^2$, where $\sigma^2 = \mathbb{E}\left[-\log \bar{P}_e^t(Y_{t+1})\right]$ is the irreducible aleatoric uncertainty. The solid orange and blue lines represent MSEs of BMA and ICL, respectively. Here, the dashed orange line corresponds to the $\sigma^2 + \epsilon_{\text{Bayes}}^t + \triangle_{\text{XS}}$, which serves as a lower bound on MSEs of ICL. The shift by $\triangle_{\text{XS}}$ induces suboptimality that requires at least $LB(q)$ additional number of demonstrations for ICL to achieve the requirement $q$, compared to BMA.

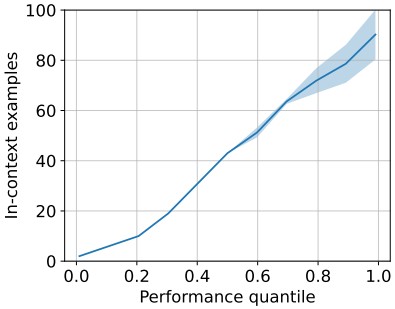

Figure A3: The number of demonstrations (y-axis) required to achieve each performance quantile (x-axis). The shaded area represents the standard error. We note that performance quantile $\mathcal{Q} = 0.6$ is achieved by $T_{\text{Train}}$ number of demonstrations on average.

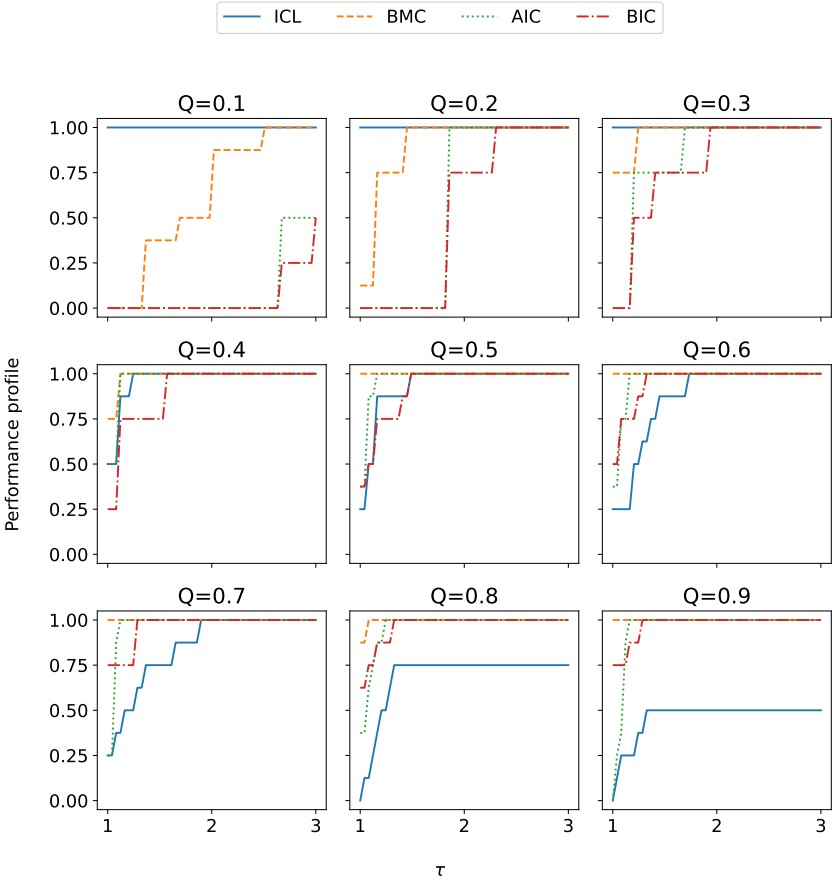

Figure A4: Performance profiles $\rho_b$ across different performance ratios $\tau$ under different target performance quantiles $\mathcal{Q}$. Each curve represents the probability that a method achieves the desired performance within a factor $\tau$ of the best method's sample complexity (x-axes).

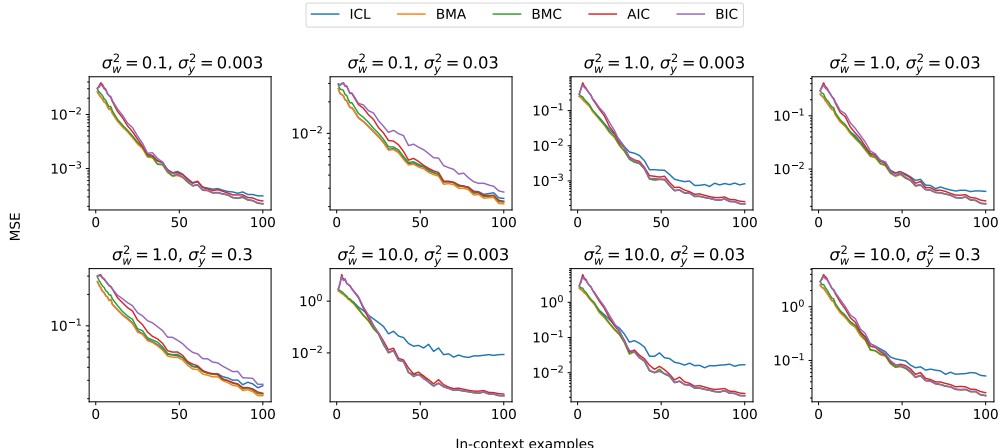

Figure A5: Mean squared errors for different demonstration sizes.

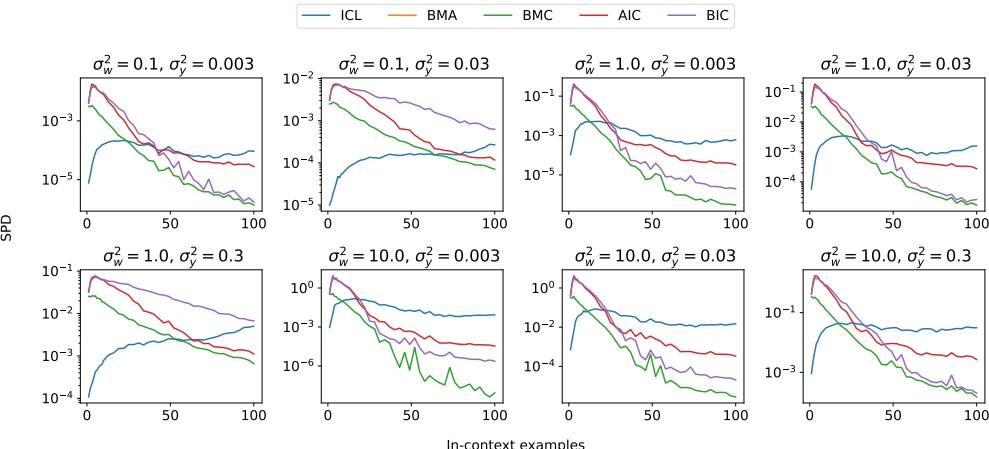

Figure A6: Squared prediction differences between BMA and other methods for different demonstration sizes.

