# OpenReview forum: "Technical Debt in In-Context Learning: Diminishing Efficiency in Long Context"
_NeurIPS.cc/2025/Conference — NeurIPS 2025 poster_

### Official Review · Reviewer_LvzK · 2025-06-10

**Clarity:** 3
**Significance:** 3
**Originality:** 3
**Rating:** 5
**Confidence:** 3

**Summary:**

This submission considers the optimality of algorithms implemented by transformers via in-context learning (ICL). The main finding is that transformers implement suboptimal ICL algorithms, especially in the limit of long contexts (ie, more in-context examples) and when the context length at test time is longer than during training (length generalization). This result is based on a _stylized_ learning problem with a hierarchical data-generating process.

**Questions:**

Raventós et al (NeurIPS 2023) argue that transformers achieve better generalization performance by _failing_ to be Bayes optimal (with respect to the empirical pretraining distribution). My understanding is that in the current submission, transformers never outperform Bayes optimal algorithms. I would appreciate a discussion on this point.

**Ethical Concerns:**

["NO or VERY MINOR ethics concerns only"]

**Final Justification:**

The authors have resolved most of my questions, and I find the topic current and interesting; therefore, I raised my rating from 4 to 5.

**Limitations:**

Please see my comments about the _stylized_ setting in **Strengths And Weaknesses** above.

**Paper Formatting Concerns:**

No paper formatting concern

**Quality:**

3

**Strengths And Weaknesses:**

Understanding ICL in transformers and its limits is timely and important. I think this paper makes a meaningful contribution to this area.

**Strength**
- The use of a toy problem allows for a more in-depth study of ICL performance
- Using principled algorithms as baselines enables quantification of ICL sample (context) efficiency
- The paper is well structured and reads rather well

**Weakness**
- The _stylized_ setting is quite specific, the claim is more general.
- I understand that the authors aim to illustrate the phenomenon, and I consider this a valuable contribution. But I think the abstract could be more informative about the learning problem considered.
- The information-theoretic analysis is interesting, but it is not directly compared to experimental results. Perhaps mine is an unpopular opinion, but I feel that the technical details do not add much to the experimental results here. I'd rather see more experimental results on different learning problems.
- The related work section is rather light. I note that the most recent reference is more than a year old (essentially an epoch in modern ML research). Expanding this section would allow the authors to better place their results in the growing body of research seeking to characterize and understand ICL with _"physics-style" or "synthetic benchmark"_ approaches.

---

> ### Author Rebuttal · Authors · 2025-07-31
>
> We thank Reviewer LvzK for the thoughtful feedback and insightful questions. We are encouraged that the reviewer found our work to be a "meaningful contribution" to the important area of understanding ICL. We address the specific weaknesses and questions below.
>
>
> ## **W1:** "The stylized setting is quite specific, the claim is more general."
>
>
> We agree with the reviewer on the importance of clarifying the scope of our claims. While our experiments are necessarily conducted with a regression problem in a stylized setting to benchmark ICL against the Bayes optimal estimator, our central claim about the diminishing sample efficiency of ICL is grounded in a general theoretical analysis that is not specific to our experimental setup.
>
> Our information theoretic analysis hinges on a general condition on a shape of the excess risk curve (Assumption 4.1) sufficient to cause the diminishing efficiency in long context. This condition is not tied to a specific model size or data type. Besides, the sufficient condition stated in Assumption 4.1, which is about the existence of lower bound of the excess risk curve, align well with empirical observations in large-scale models. For instance, Anil et al. (2022) and Zhou et al. (2024) show that state-of-the-art LLMs struggle when the context length at test time significantly exceeds that seen during training. These challenges transformers face with length generalization prove the existence of the excess risk’s lower bound in the length generalization regime.
>
> Therefore, while explicitly confirming the phenomenon in state-of-the-art LLMs remains a valuable future direction, the alignment of our results with recent large-scale studies strongly supports the practical relevance and generalizability of our findings.
>
>
> (Line 317) “We remark that the deficiencies of state-of-the-art LLMs beyond length generalization regime (Anil et al., 2022; Zhou et al., 2024) prove the existence of the excess risk’s lower bound within the length generalization regime.”
>
>
> Anil et al. (2022). Exploring length generalization in large language models. In NeurIPS.
>
> Zhou et al. (2024). Transformers can achieve length generalization but not robustly. arXiv.
>
>
> ## **W2:** "I understand that the authors aim to illustrate the phenomenon, and I consider this a valuable contribution. But I think the abstract could be more informative about the learning problem considered."
>
> Thank you for this valuable suggestion. To provide readers with a clearer problem setup and the complexity involved in the learning task, we have revised the abstract (Lines 6-7) as follows:
>
>
> “To investigate this, we adopt a meta ICL setup where each prompt corresponds to a regression problem, with the target function sampled from a hierarchical distribution, requiring inference over both latent model class and task parameters. In this setting, we benchmark sample complexity of ICL against principled learning algorithms, including the Bayes optimal estimator, under diverse performance requirements.”
>
>
>
>
> ## **W3:** "The information-theoretic analysis is interesting, but it is not directly compared to experimental results. Perhaps mine is an unpopular opinion, but I feel that the technical details do not add much to the experimental results here. I'd rather see more experimental results on different learning problems. "
>
> We appreciate this insightful comment. While more experiments on diverse tasks would certainly add empirical breadth, our theoretical analysis in Section 4 provides a unique and vital contribution that experiments alone cannot.
> The theory does not just support the experimental findings; it provides a causal explanation for them. It demonstrates that the observed diminishing sample efficiency is not an artifact of our specific model, task, or training procedure, but rather an inherent limitation of ICL mechanism when faced with a large number of demonstrations. By identifying the formal conditions under which this inefficiency occurs, our theory correctly attributes the phenomenon to the ICL mechanism itself. We believe this provides a more fundamental insight than additional experiments on other tasks would.
>
>
> To clarify this crucial role of our theory, we have added the following sentence to the manuscript (Line 276):
>
>
> “This theoretical grounding is critical, as it establishes that the diminishing inefficiency is an intrinsic property of the ICL mechanism, rather than an artifact of a particular experimental setup. This correct attribution is essential for guiding future work aimed at overcoming this limitation.”
>
>
> ## **W4:** "The related work section is rather light. I note that the most recent reference is more than a year old (essentially an epoch in modern ML research). Expanding this section would allow the authors to better place their results in the growing body of research seeking to characterize and understand ICL with "physics-style" or "synthetic benchmark" approaches."
> ​​
>
> Thank you for the great suggestion! We agree that contextualizing our work within the rapidly evolving literature is essential. We have updated our related work section to include several relevant recent papers that adopt similar stylized settings to understanding ICL.
>
>
> (Line 400): “More recently, these stylized settings have been used to probe other sophisticated behaviors of ICL. This includes analyzing transformers' in-context model selection and preference for simpler hypotheses (Deora et al., 2024; Elmoznino et al., 2025), their ability to infer causal structures (D’Angelo et al., 2025), and the implicit connection between ICL and low-rank updates to MLP layers (Dherin et al., 2024).”
>
>
> (Revised Line 402) This new perspective unveils a unique insight: the fundamental inefficiency of ICL in the many-shot learning regime, a critical limitation that was not the focus of these prior analyses.
>
>
> Deora et al. (2025). In-Context Occam's Razor: How Transformers Prefer Simpler Hypotheses on the Fly. arXiv.
>
> D'Angelo et al. (2025). Selective induction heads: How transformers select causal structures in context. In ICLR.
>
> Elmoznino et al. (2025). In-context learning and Occam's razor. In ICML.
>
> Dherin et al. (2025). Learning without training: The implicit dynamics of in-context learning. arXiv.
>
>
> ## **Q1:** "Raventós et al (NeurIPS 2023) argue that transformers achieve better generalization performance by failing to be Bayes optimal (with respect to the empirical pretraining distribution). My understanding is that in the current submission, transformers never outperform Bayes optimal algorithms. I would appreciate a discussion on this point. "
>
> We appreciate the reviewer’s important point and clarify the difference between the notion of Bayes optimality in our work and in Raventos et al. (2023).
> - Raventós et al. (2023): Optimality w.r.t. the finite pretraining data. They show that as pretraining data diversity increases, transformers deviate from the Bayes-optimal predictor defined on that finite pretraining set. This deviation actually improves generalization because it moves the transformer’s behavior closer to a ridge estimator that is optimal for the true underlying data distribution.
> - Our work: Optimality w.r.t. the true data-generating distribution. We define the Bayes-optimal estimator with respect to the ground-truth hierarchical distribution from which tasks are sampled. By definition, no algorithm can achieve a lower average risk on this distribution. For a side note, the notion of Bayes optimality here is the same as in the optimality of the ridge estimator in Raventós et al. (2023).
>
>
> In short, Raventós et al. show that transformers are not optimal with respect to their limited training data, which allows them to be more optimal with respect to the true data source. Our work compares transformers to the theoretical ceiling of performance—the true Bayes-optimal estimator. Thus, there is no contradiction: transformers never outperform our Bayes-optimal baseline, which is consistent with the findings of Raventós et al. when properly interpreted.
>
> To clarify this in the paper, we have revised Line 118:
>
> “The Bayes-optimal estimator defined in (2) minimizes the expected risk with respect to the true hierarchical data-generating distribution. This is distinct from the notion of optimality with respect to an empirical pretraining distribution with finite samples, as considered in Raventós et al. (2023), where deviating from empirical optimality can improve generalization.”
>
>
> We hope these revisions and clarifications have fully addressed the reviewer's concerns. We thank the reviewer again for the constructive engagement.

---

> > ### Comment · Reviewer_LvzK · 2025-08-03
> >
> > Thank you for your detailed reply, which has addressed most of my questions, and I will raise my rating. I have a couple of follow-up questions.
> >
> > *Efficiency of learning algorithm.*  Thank you for clarifying the scope and the goals of the information-theoretic analysis. I find this topic around algorithmic optimality quite interesting, and I wonder whether and how this analysis relates to the information efficiency of learning algorithms studied in arXiv:2208.03848.
> >
> > *On theoretical analysis.* I appreciate the insights from the analysis. And as far as I can tell, the theoretical results support the experimental findings, albeit qualitatively. This finding is valuable, but I am curious whether it can be made even more convincing. Is there no way to have a quantitative comparison between theory and experiments, even for a simpler experimental setting?

---

> > > ### Author Response · Authors · 2025-08-05
> > >
> > > We thank Reviewer LvzK for their insightful follow-up questions and for indicating an improved rating. We are grateful that our detailed rebuttal addressed your main concerns. Below we provide responses to the follow-up questions.
> > >
> > > # Q1: Efficiency of learning algorithm.
> > >
> > >
> > > Thank you for raising this thought-provoking question. You've highlighted an interesting conceptual link between our work and Ngampruetikorn & Schwab (2022). While the problem settings and goals of analyses differ, both frameworks investigate the efficiency of a learning algorithm relative to an optimal benchmark. The core distinction lies in how "efficiency" is defined and measured:
> > >
> > > Ngampruetikorn & Schwab (2022) focus on information efficiency of learning algorithms (Gibbs posterior and ridge regression) through the information bottleneck framework (Tishby et al., 1998). They analyze how much "irrelevant" information or coding redundancy the learner encodes to achieve a certain performance level. With this view, they analyze impacts of a coefficient of the Tikhonov regularization, randomness of the Gibbs posterior, and overparameterization on the efficiency of the Gibbs posterior.
> > >
> > > In contrast, we study the sample complexity. We analyze how many “excess” demonstrations in-context learning require compared to the Bayes optimal estimator to reach a certain performance level. Our central finding is that the inefficiency of ICL increases with the target performance, leading to the observation “diminishing efficiency of ICL in long context.”
> > >
> > > Despite these differences perspectives, both frameworks aim to quantify suboptimality of learning algorithms either in terms of encoded irrelevant information (Ngampruetikorn & Schwab) or excess number of required demonstrations given some performance requirements. This simple and clear connection suggests a potentially unifying perspective on algorithmic efficiency. We are grateful to the reviewer for pointing this out and will add a discussion to the revised manuscript to make this relationship explicit.
> > >
> > > Ngampruetikorn & Schwab (2022). Information bottleneck theory of high-dimensional regression: relevancy, efficiency and optimality. In NeurIPS.
> > >
> > > Tishby et al. (1998)   The information bottleneck method. In Allerton Conference on Communication, Control and Computing.
> > >
> > >
> > >
> > > # Q2: On theoretical analysis.
> > >
> > >
> > > This is again an insightful question from Reviewer LvzK. At present, a direct quantitative comparison between our theoretical results and experimental results is unfortunately not possible, because our analysis does not yield an exact (asymptotic) rate of excess risk or sample complexity. This is because our theory is designed to establish the existence of regimes where ICL becomes sample-inefficient compared to the Bayes-optimal estimator, rather than to predict precisely how suboptimal it is.
> > >
> > >
> > > We want to highlight two main challenges to achieve the quantitative prediction: first, the sample complexity of the Bayes-optimal estimator is problem-dependent and often requires strong assumptions about the data-generating process. Further, the exact rate at which ICL's excess risk decays as a function of the number of demonstrations is a significant analytical challenge in its own right. We view this as an important open problem and an exciting direction for future work, and leave it as a future work in the conclusion of our revised manuscript.

---

### Official Review · Reviewer_N11X · 2025-07-01

**Clarity:** 3
**Significance:** 3
**Originality:** 2
**Rating:** 5
**Confidence:** 3

**Summary:**

This work investigates the diminishing efficiency of in-context learning (ICL) in the many-shot scenario. The authors pre-train a Transformer model on a given task distribution for ICL and compare its sample efficiency against optimal baselines. They discover that the efficiency of in-context learning is high in the few-shot regime and decreases as the context length increases. They justify their observations by analyzing the suboptimality of ICL and showing that the suboptimality increases as the demonstration size grows.

**Questions:**

I see the authors use the Transformer as their default model architecture to examine the drop in sample efficiency for in-context learning with long contexts. I wonder what special properties of the Transformer they leveraged for their analysis? I don't see any architectural requirement in their assumptions. Thus, why don't the authors also consider other sequence modelling models, such as the LSTM, as ICL is not exclusive to the Transformer?

**Ethical Concerns:**

["NO or VERY MINOR ethics concerns only"]

**Final Justification:**

- This paper studies an important risk of in-context learning.
- The authors' rebuttal addressed my concerns regarding the scalability of the phenomenon to larger models, the excess risk lower bound, and the scope of the work.
- In response to my question, the authors conducted additional experiments to study the diminishing efficiency of in-context learning with models other than the Transformer.

I therefore keep my decision to accept the paper.

**Limitations:**

1. I think the authors should also acknowledge that their model size and training data diversity are not comparable to today's LLMs.
2. I believe the authors should also acknowledge that they only investigated in-context regression. Otherwise, they should explicitly state that they use the term in-context learning to indicate in-context regression specifically.

**Paper Formatting Concerns:**

No formatting concerns.

**Quality:**

3

**Strengths And Weaknesses:**

**Strengths**
- The paper is overall clearly written. The notations are well-defined, and intuitions are provided to interpret the results.
- The empirical study is well-designed with good statistical rigour.
- The diminishing efficiency phenomenon in ICL is a real and practical concern for ICL applications.

**Weaknesses**
- The authors draw connections between their results and large language models. However, it is unclear whether the decreased efficiency of ICL in their particular setup and model size persists in LLMs, where the models are significantly larger and trained on a wide range of demonstration sizes. In the authors' setup, they only considered a fixed largest demonstration size $T_{train}$.
- The analysis is based on the Assumption that there exists a uniform lower bound on the excess risk after a certain demonstration length, which can be too strong to hold.
- This work only considered in-context supervised regression problems, leaving classification problems and other learning paradigms, such as reinforcement learning, unexamined.

---

> ### Author Rebuttal · Authors · 2025-07-31
>
> We sincerely thank Reviewer N11X for the positive assessment and insightful, constructive feedback. We have revised our manuscript to incorporate the excellent suggestions, which we believe have strengthened the paper.
>
> ## **W1:** “The authors draw connections between their results and large language models. However, it is unclear whether the decreased efficiency of ICL in their particular setup and model size persists in LLMs, where the models are significantly larger and trained on a wide range of demonstration sizes. In the authors' setup, they only considered a fixed largest demonstration size”
>
> This is a crucial point. While our experiments use a controlled setup, we argue that the phenomenon we identify—diminishing ICL efficiency with long context—is a fundamental issue likely to generalize to large-scale models. Our reasoning is twofold:
>
> - General Theoretical Foundation: Our theoretical analysis in Section 4 is formulated with a general setting. The core finding relies on Assumption 4.1, which is not tied to a specific model size or data distribution. As a result, the theoretical insights we offer remain broadly applicable, transcending the specifics of our experimental setup.
> - Alignment with Large-Scale Empirical Evidence: The practical consequence of our theoretical finding is that Assumption 4.1 is a sufficient condition for the diminishing efficiency in long context. Crucially, Assumption 4.1 generality aligns well with empirical observations from recent large-scale transformer studies. For instance, Anil et al. (2022) and Zhou et al. (2024) both report that even state-of-the-art LLMs struggle to extrapolate when test-time context length exceeds that seen during training. Given that the Bayes risk improves with more demonstrations, this means that the excess risk curve is deteriorating in the extrapolating regime.
>
>
> Therefore, while explicitly confirming the phenomenon in state-of-the-art LLMs remains a valuable future direction, the alignment of our results with recent large-scale studies strongly supports the practical relevance and generalizability of our findings.
>
> Anil et al. (2022). Exploring length generalization in large language models. In NeurIPS.
>
> Zhou et al. (2024). Transformers can achieve length generalization but not robustly. arXiv.
>
>
> ## **W2:** "The analysis is based on the Assumption that there exists a uniform lower bound on the excess risk after a certain demonstration length, which can be too strong to hold."
>
>
> We thank the reviewer for highlighting this critical detail, and we have revised the manuscript to clarify the scope of Assumption 4.1. The assumption is not intended to be a single universal bound that holds for all models and tasks. Rather, it posits that for a given transformer and a given task environment, a lower bound of the excess risk exists (especially when generalizing to the demonstration size significantly larger than that of used in training).
>
> This model-and-environment-specific assumption is strongly supported by empirical evidence. We observe this performance plateau directly in our own experiments (Figures 3 and 4). Furthermore, this behavior aligns well with empirical observations in large-scale models. For instance, Anil et al. (2022) and Zhou et al. (2024) show that state-of-the-art LLMs struggle when the context length at test time significantly exceeds that seen during training. These challenges transformers face with length generalization prove the existence of the excess risk’s lower bound in the length generalization regime.
>
> To ensure this is clear, we have explicitly revised the statement of the assumption in the manuscript to be model and environment-dependent (Line 314):
>
>
> “For an environment $\mathcal{E}$ and a transformer $TF_\theta$, there exist constants  … ”
>
>
> ## **W3:** “This work only considered in-context supervised regression problems, leaving classification problems and other learning paradigms, such as reinforcement learning, unexamined.”
>
>
> This is a correct observation, and we thank the reviewer for suggesting we make our scope and its justification clearer. We have now explicitly addressed the focus on regression and other limitations in the revised manuscript.
>
>
> Our focus on regression is a deliberate methodological choice. This paradigm allows for the analytical derivation of the Bayes-optimal performance, providing a "gold standard" benchmark. This is essential for our goal of rigorously and quantitatively measuring the sample efficiency of ICL relative to a theoretical optimum. While extending this analysis to classification or RL is an exciting future direction, it poses challenges in analytically deriving the optimal baselines.
>
>
> To improve transparency, we have made the following additions to the manuscript:
>
>
>
> (Line 6) “To investigate this, we adopt a meta ICL setup where each prompt corresponds to a regression problem, with the target function sampled from a hierarchical distribution, requiring inference over both latent model class and task parameters. In this setting, we benchmark sample complexity of ICL against principled learning algorithms, including the Bayes optimal estimator, under diverse performance requirements.”
>
>
> (Line 81) “Following common practice in the meta ICL literature, we focus on regression problems for which the Bayes-optimal performance can be derived analytically, allowing us to precisely quantify the sample complexity of in-context learning and principled learning algorithms.”
>
>
> (Line 554) “Further, the model size and training data diversity used in our meta ICL setup are significantly smaller and less diverse than those used in modern LLMs, such as GPT-4 and Gemini 2.5, which typically possess hundreds of billions or even trillions of parameters trained on vast, heterogeneous datasets.”
>
>
>
> ## **Q1:** “I see the authors use the Transformer as their default model architecture to examine the drop in sample efficiency for in-context learning with long contexts. I wonder what special properties of the Transformer they leveraged for their analysis? I don't see any architectural requirement in their assumptions. Thus, why don't the authors also consider other sequence modelling models, such as the LSTM, as ICL is not exclusive to the Transformer?”
>
>
>
> This is an excellent question that prompted us to conduct new experiments to strengthen our claims. Our initial focus on the Transformer was due to its dominance in ICL research and state-of-the-art large language models. However, the core of our theoretical analysis is not specific to the attention mechanism but rather to the general properties of sequence models, as long as it satisfies the conditions of our analysis (e.g., Assumption 4.1).
>
>
> Motivated by the reviewer's suggestion, we ran additional experiments with LSTM and Mamba architectures. The results are highly informative: The Transformer consistently outperformed both LSTM and Mamba in overall ICL performance, aligning with recent comparative studies (Akyürek et al., 2024; Lee et al., 2024). Crucially, the central phenomenon of diminishing sample efficiency in long context persisted across all three architectures. This finding is also consistent with recent work analyzing the length extrapolation limits of Mamba (Ben-Kish et al., 2025). These new findings strongly reinforce our central claim: the diminishing efficiency of ICL in long context is a general property of the ICL paradigm in sequence models having a non-vanishing excess risk, not an artifact of the Transformer architecture. This underscores the broad applicability of our theoretical framework. We will add a discussion of these results to the revised manuscript.
>
>
> Akyürek et al. (2024). In-context language learning: Architectures and algorithms. arXiv
>
> Lee et al.  (2024). Is attention required for icl? exploring the relationship between model architecture and in-context learning ability. ICLR
>
> Ben-Kish et al. (2025) Decimamba: Exploring the length extrapolation potential of mamba. In ICLR.
>
>
> **Limitations:** See the responses to W3 above.
>
>
> We thank the reviewer again for the insightful suggestions, which have significantly improved the rigor and clarity of our work.

---

> > ### Comment · Reviewer_N11X · 2025-08-05
> >
> > I thank the authors for addressing my concerns and providing further clarification. I look forward to reading the new empirical results in the next version of the manuscript. Since I have already voted for the paper's acceptance, I am keeping my rating as is.

---

> > > ### Author Response · Authors · 2025-08-05
> > >
> > > Thank you for your valuable feedback and continued support. We are pleased that our clarifications were helpful and will certainly include the new empirical results in the revised manuscript.

---

### Official Review · Reviewer_pLhk · 2025-07-01

**Clarity:** 2
**Significance:** 1
**Originality:** 1
**Rating:** 2
**Confidence:** 4

**Summary:**

This paper investigates the sample-efficiency of in-context learning (ICL) in transformer models, focusing on how many demonstrations are needed to reach a target prediction error compared to Bayes-optimal or principled learners. Using a stylized regression benchmark with Fourier-series functions and controlled noise, the authors show that transformers are nearly optimal in the few-shot regime but become significantly less efficient as the number of in-context examples increases. They formalize this observation using a performance-profile metric and support it with an information-theoretic analysis, which proves that under a certain assumptions the transformer’s efficiency must diverge from the Bayes rate in the long-context limit.

**Questions:**

See weakness.

**Ethical Concerns:**

["NO or VERY MINOR ethics concerns only"]

**Limitations:**

yes.

**Quality:**

2

**Strengths And Weaknesses:**

Strengths:

1. The paper is addressing an interesting problem that what you lose if you use ICL instead of principled algorithms.


Weaknesses

1. From expressivity point of view the transformer can implement many principled algorithm in its forward pass, an idea popularized by Oswald et al.[1] and many other follow-ups. So, if that's the case then I do not see the central point of this paper. The context length for fixed error level should be exactly the same for transformer or base algorithm.

2. Related to 1, ICL is nothing special from traditional learning paradigm. We have some demonstration and we want to predict a new query, so transformer can be thought as an algorithm potentially.

3. I think the main problem of this paper is Assumption 4.1,  basically as the model gets larger and more expressive, and also number of pre-traing increases we can get closer and closer to best possible error. And this is align with scaling-law common wisdom.

4. Related to 3,  because the transformer is meta-trained across many tasks, it can, in principle, exploit cross-task regularities that a hand-crafted Bayes estimator (tuned per task) cannot access. The analysis ignores this potential advantage.

Bottom line: I think this paper assumption is too restrictive and unrealistic, and not adding any new insight to what ICL actually is.

[1] [Transformers learn in-context by gradient descent](https://arxiv.org/abs/2309.14666)
Johannes von Oswald, Eyvind Niklasson, Ettore Randazzo, João Sacramento, Alexander Mordvintsev, Andrey Zhmoginov, Max Vladymyrov. *arXiv preprint arXiv:2309.14666*, 2023.

---

> ### Author Rebuttal · Authors · 2025-07-31
>
> We thank Reviewer pLhk for their time and effort in reviewing our manuscript and providing constructive feedback. Below, we carefully address the reviewer’s comments, clarify certain misconceptions, and emphasize the novelty and significance of our findings with support from recent relevant studies.
>
>
> ## **W1:** "From expressivity point of view the transformer can implement many principled algorithm in its forward pass, an idea popularized by Oswald et al.[1] and many other follow-ups. So, if that's the case then I do not see the central point of this paper. The context length for fixed error level should be exactly the same for transformer or base algorithm."
>
>
> We agree that transformers, in theory, have the capacity to implement principled learning algorithms in their forward pass (von Oswald et al., 2023). However, the crucial distinction—and the central point of our paper—lies between this theoretical potential and the algorithms that are actually learned through standard meta-training. The work by von Oswald et al. is an existence proof, showing that a specific weight configuration can mimic an algorithm like gradient descent; it does not guarantee that pre-training will converge to this or any other optimal implementation.
>
> Our work investigates this exact gap between theory and practice. We ask: What kind of algorithm do transformers learn, and what are its limitations? Our key finding is that while the learned algorithm appears near-optimal in the few-shot regime (a finding consistent with previous works, e.g., Garg et al. (2022) and Panwar et al. (2024)), its sample efficiency degrades significantly in the many-shot regime. This reveals a critical limitation of what is learned in practice, which is the novel insight of our work.
>
> This also clarifies the reviewer's comment on sample complexity. If a transformer did perfectly implement the Bayes-optimal algorithm, its sample complexity would indeed be the same. But since our findings show it implements a suboptimal algorithm, its sample complexity is naturally different and, as we demonstrate, worse.
> To clarify this in the manuscript, we have added the following:
> (Line 54) "While transformers are theoretically capable of implementing principled algorithms (von Oswald et al., 2023), we show that ICL, actually learn by training, deviates significantly from optimality in the many-shot regime.”
>
> Garg et al. (2022) What can transformers learn in-context? a case study of simple function classes. In NeurIPS.
>
> Panwar et al. (2024)  In-context learning through the Bayesian prism. In ICLR
>
>
> ## **W2:** "Related to 1, ICL is nothing special from the traditional learning paradigm. We have some demonstration and we want to predict a new query, so the transformer can be thought of as an algorithm potentially."
>
> We respectfully disagree with the characterization that "ICL is nothing special." While ICL can be viewed as an algorithm, its underlying mechanism is fundamentally distinct from traditional learning paradigms, and this distinction is critical to our paper's contribution.
>
> Traditional supervised learning involves updating a model's explicit parameters (weights) via an optimization process like gradient descent. In contrast, ICL performs learning purely through inference within a fixed-weight network. The "learning" happens as information provided in the demonstrations is processed without any weight updates.
>
> This mechanical distinction is not trivial; it imposes unique constraints that do not exist in the same way for traditional learners. Investigating the properties and limitations of this unique inference-based learning mechanism is an active and important area of research (e.g., Brown et al. 2020; Xie et al., 2022; Akyürek et al., 2023; Jeon et al., 2024; Elmoznino et al., 2025). Our work contributes directly to this by providing a new, formal understanding of ICL's sample efficiency in long context.
>
> Brown et al. (2020) Language models are few-shot learners. In NeurIPS.
>
> Xie et al. (2022) An explanation of in-context learning as implicit Bayesian inference. In ICLR.
>
> Akyürek et al. (2023). What learning algorithm is in-context learning? investigations with linear models. In ICLR.
>
> Jeon et al. (2024) An information-theoretic analysis of in-context learning. In ICML.
>
> Elmoznino et al. (2025). In-context learning and Occam's razor. In ICML.
>
>
> ## **W3:** "I think the main problem of this paper is Assumption 4.1, basically as the model gets larger and more expressive, and also the number of pre-training increases we can get closer and closer to the best possible error. And this is aligned with scaling-law common wisdom."
>
>
> We thank the reviewer for raising this point, as it allows us to clarify the scope and basis of Assumption 4.1. Reviewer pLhk’s concern appears to stem from a misunderstanding, connecting our assumption to scaling laws.
>
>
> Our assumption is not about performance scaling with model size or pre-training data. Instead, Assumption 4.1 simply states that the excess risk curve is lower bounded when it confronts a sufficiently long context exceeding the context length during pretraining. Far from being "restrictive or unrealistic," this is a robust and widely observed empirical phenomenon. It is a core finding in the literature on length-generalization limitations in transformers (Anil et al., 2022; Zhou et al., 2024) and is a pattern we explicitly validate in our own experiments (Figures 3 and 4). Therefore, we think that Assumption 4.1 is a reasonable, realistic, and empirically grounded foundation for our theoretical analysis of why ICL's efficiency degrades in the many-shot regime.
>
>
> Anil et al. (2022). Exploring length generalization in large language models. In NeurIPS.
>
> Zhou et al. (2024). Transformers can achieve length generalization but not robustly. arXiv.
>
>
> ## **W4:** "Related to 3, because the transformer is meta-trained across many tasks, it can, in principle, exploit cross-task regularities that a hand-crafted Bayes estimator (tuned per task) cannot access. The analysis ignores this potential advantage."
>
>
> As Reviewer pLhk pointed out, transformers' meta-learning capabilities allow them to exploit cross-task regularities, potentially offering advantages over handcrafted estimators learned for each task.
>
>
> First, our empirical setup explicitly accounts for this advantage. We pre-train the transformer on a diverse, hierarchical distribution of tasks. The ICL performance we report is therefore the result of the model already having exploited these cross-task regularities.
>
>
> Second, and more fundamentally, Reviewer pLhk might misunderstand the role of the Bayes-optimal estimator in the benchmark. The Bayes-optimal estimator is not a "hand-crafted Bayes estimator (tuned per task)." It is a theoretical performance ceiling derived analytically from the true, known data-generating distribution (cf. Eq. 2). By definition, no algorithm, whether it is a transformer using meta-learning or any other method, can achieve a lower expected risk on this distribution.
>
>
> Therefore, our comparison is fair: we compare the transformer's meta-learned performance against the theoretical best-case performance on the same task distribution. The fact that the transformer falls short, particularly in the many-shot regime, is a significant and non-obvious finding.
>
>
> We hope this clarifies that our assumptions are well-grounded and our findings provide a novel and important insight into the fundamental limitations of in-context learning.

---

> ### Comment · Reviewer_pLhk · 2025-08-06
>
> I thank the authors for the clarification. However, my concerns are still not addressed:
>
> About W1, W2: What I meant by saying that ICL is not special is that, at test time after the pre-training phase, it can be viewed as implementing a fixed algorithm that consumes the in-context demonstrations and returns an estimate for the query. How far this algorithm falls from the optimal one depends on several factors, including:
>
> 1. The training dynamics of transformers
>
> 2. The number of pre-training samples (e.g., in the hypothetical extreme of training on all possible samples, one would expect optimal performance)
>
> 3. Context length
>
> I agree that context length matters, but transformers are highly expressive (even a two-layer MLP is quite expressive), so in theory they can implement any algorithm. If they do not, it must be due to either an insufficient number of pre-training samples or limitations in the training procedure. Hence, I do not understand the abstract’s claim: “Through an information-theoretic analysis, we show that the diminishing efficiency is inherent to ICL.” Why is it inherent?
>
> W3: I still do not see why Assumption 4.1 is reasonable. Length generalization issues may simply reflect limitations of the training dynamics. If a transformer could implement the optimal algorithm (acknowledging this is unrealistic), it should generalize perfectly to longer contexts. This is therefore a limitation of the training process, not of ICL itself.
>
> In addition, although your experimental setting has been extensively studied in prior work, it is far from the realistic ICL observed in LLMs. In this setup, the models are deliberately trained on explicit ICL tasks, unlike the natural emergence of ICL, making the setting less relevant.
>
> Overall, I remain unconvinced by the paper’s main message and take-aways and thus keep my score unchanged.

---

> > ### Author Response · Authors · 2025-08-09
> > **Author response (1/2)**
> >
> > We thank Reviewer pLhk for the follow-up and for clarifying their earlier points. We have incorporated their valuable feedback, which has further strengthened the manuscript. Our detailed responses below address the specific points raised.
> >
> >
> > # W1, W2
> >
> > We appreciate Reviewer pLhk’s elaboration on the claim. However, we believe the current reasoning still conflates two distinct notions: (1) what a transformer can represent in principle, and (2) what it actually learns through (pre)training. We address this distinction first and then explain how our information-theoretic analysis shows that diminishing efficiency in long contexts is an inherent property of ICL under mild assumptions.
> >
> > The reviewer suggests that because transformers are highly expressive, they should, once successfully trained, implement the optimal algorithm in their forward pass. While it is true that transformers are capable of representing optimal algorithms in principle (von Oswald et al., 2023), this is an existence proof and does not imply that pretraining will converge to such weight configurations without strong assumptions (as discussed in Lines 28-30). As noted in our previous response, our empirical results indicate that “while the learned algorithm appears near-optimal in the few-shot regime, its sample efficiency degrades significantly in the many-shot regime.” We also draw attention to the newly added clarification in Line 54:
> >
> > “While transformers are theoretically capable of implementing principled algorithms (von Oswald et al., 2023), we show that ICL, as actually learned through pretraining, deviates significantly from optimality in the many-shot regime.”
> >
> >
> >
> >
> > __Why is diminishing efficiency in long context inherent to ICL?__
> >
> >
> >
> >
> > We agree with the reviewer's premise: if a transformer could be trained to perfectly implement the optimal algorithm, this issue would vanish. The core of our argument, however, is that the diminishing efficiency is inherent to the ICL mechanism itself, given the reality of imperfectly trained models.
> >
> > Specifically, our argument proceeds with two steps:
> > 1. Empirical premise (Assumption 4.1). Real-world pretrained transformers are not perfect. For sufficiently long contexts, they exhibit a persistent, non-zero excess risk as observed in our work (Figures 3 & 4) and prior studies (Anil et al., 2022; Zhou et al., 2024). See our response to W3 about why this is a mild assumption.
> > 2. Theoretical consequences (Theorems 4.2 and 4.3). Given this empirical premise, our information-theoretic analysis shows that the ICL mechanism will necessarily suffer from diminishing sample efficiency as the demonstration size increases.
> >
> >
> >
> >
> > Conversely, it is worth noting that the very same pretrained model is not subject to this diminishing efficiency when its weights are fine-tuned by gradient-based optimization methods, as it then becomes an asymptotically efficient predictor. This contrast highlights that the diminishing efficiency we observe is an inherent limitation of ICL as an adaptation mechanism without parameter updates. We will add a clarification to the paper to make this specific meaning of "inherent" more explicit.
> >
> >
> >
> >
> > Anil et al. (2022). Exploring length generalization in large language models. In NeurIPS.
> >
> >
> > Zhou et al. (2024). Transformers can achieve length generalization but not robustly. arXiv.

---

> > ### Author Response · Authors · 2025-08-09
> > **Author response (2/2)**
> >
> > # W3
> > We are glad that the earlier conflation between scaling-law behavior and our assumption on the existence of a lower bound appears resolved.
> >
> >
> > The reviewer’s new comment concerns that the lower bound in Assumption 4.1 would vanish if a transformer generalized perfectly to longer contexts, and otherwise reflects a limitation of the training process. We believe this perspective hinges on two misunderstandings:
> >
> >
> >
> >
> > __1. On the role of Assumption 4.1.__
> > Assumption 4.1 is descriptive, not causal. As defined in Line 314, it formalizes the existence of the lower bound:
> > There exist constants $(\bar{t}, \triangle_{\text{XS}}) \in (\mathbb{N}, \mathbb{R}{+})$ such that $0 \leq \triangle{\text{XS}} \leq \epsilon_{\text{XS}}^{t^\prime}$ for all $t^\prime \geq \bar{t}$.
> >
> >
> > Since our assumption does not attribute the existence of the lower bound to any particular cause, the reviewer’s suggestion that it could result from “a limitation of the training process” does not contradict Assumption 4.1 in any way.
> >
> >
> >
> >
> >
> >
> >
> >
> >
> >
> > __2. On the “perfect generalization” scenario suggested by the reviewer.__
> > As Reviewer pLhk suggested, if a transformer generalized perfectly to any context length, Assumption 4.1 does not hold. However, this is a much stronger and less realistic assumption than the imperfect generalization in long context we posit. Our assumption reflects the empirically observed phenomenon that for sufficiently long contexts, excess risk remains lower-bounded (Figures 3 and 4; Anil et al., 2022; Zhou et al., 2024).
> >
> >
> > In light of the reviewer’s comment, we have clarified this in Line 315:
> >
> >
> > “The assumption states that, after some reference point $\bar{t}$, the excess risks of $TF_\theta$ can be lower bounded as observed in Figures 3 and 4 and in empirical studies (Anil et al., 2022; Zhou et al., 2024). This may occur for various reasons such as insufficient pretraining data or intrinsic properties of architectures. Importantly, we do not assume why this happens, only that the lower bound exists.”
> >
> >
> > # On the experimental setting
> >
> >
> > We understand the concern that our controlled setting differs from the naturally emergent ICL observed in LLMs. As discussed in Section 1 (Lines 36–39) and Appendix A.1 (Lines 540–563), stylized setups are necessary for benchmarking against the Bayes optimal estimator, which is an infeasible task in real-world datasets.
> >
> >
> > We also remark that our setting generalizes the standard linear-regression meta-ICL framework (e.g., Garg et al., 2022; Raventos et al., 2023) by considering a hierarchical target function class with a latent feature space. This incorporates a rich class of functions as a basis of square-integrable functions on bounded intervals (Line 96) and implicitly requires model selection due to the latent nature of the feature space (Line 84). This yields a new finding that ICL diverges from the Bayes optimal estimator in many-shot regimes, which has not been captured in earlier, simpler setups.
> >
> >
> > Lastly, our main result concerns pessimistic behavior on the persistent gap between ICL and the Bayes optimal estimator even in the simplified setting. Given that the direct pretraining of ICL objective (our setting) yields better ICL performance than emergent ICL by eliminating the distribution shifts (Zhang et al., 2025), the real-world gap is plausibly larger than what we observe here. To highlight this connection, we have added the following paragraph in conclusion:
> >
> >
> > “The meta-ICL setup studied here is not equivalent to autoregressive pretraining of LLMs. In theory, naturally emergent ICL from autoregressive pretraining would underperform direct ICL training (1) due to distribution shift (Zhang et al., 2025). We thus conjecture that the diminishing efficiency of ICL compared to the Bayes optimal estimator in long context would be even more severe for real-world LLMs. Confirming and quantifying this phenomenon with state-of-the-art LLMs is an important direction for future work.”
> >
> >
> > Garg et al. (2022) What can transformers learn in-context? a case study of simple function classes. In NeurIPS.
> >
> > Raventos et al. (2023) Pretraining task diversity and the emergence of non-Bayesian in context learning for regression. In NeurIPS.
> >
> > Zhang et al. (2025) What and How does In-Context Learning Learn? Bayesian Model Averaging, Parameterization, and Generalization. In AISTATS.

---

### Official Review · Reviewer_kMFD · 2025-07-02

**Clarity:** 4
**Significance:** 3
**Originality:** 3
**Rating:** 5
**Confidence:** 4

**Summary:**

This paper investigates the efficiency of in-context learning (ICL) in Transformers compared to principled learning algorithms. This paper introduces a framework based on performance profiles and sample complexity to quantify the (sub-)optimality of ICL. This paper finds that while ICL is nearly as efficient as a Bayes optimal estimator in few-shot settings, its efficiency significantly degrades in many-shot scenarios that require a long context. Through an information-theoretic analysis, the paper argues that this diminishing efficiency is an intrinsic property of the ICL mechanism, caused by a non-vanishing excess risk. The work concludes that ICL carries a "technical debt" of sample inefficiency in high-performance regimes.

**Questions:**

The core of the theoretical argument rests on the non-vanishing excess risk ($epsilon_{XS}^t$). Your experiments (Fig 3b, Fig 4) convincingly show this behavior. Do you have a hypothesis for why the Transformer architecture, trained with this objective, leads to a non-vanishing excess risk? Is it related to the fixed-capacity nature of the attention mechanism or something else?

You conclude by motivating **a new generation of 'on-the-fly' adaptive methods without the diminishing efficiency.**  Could you elaborate on what such methods might look like?

**Ethical Concerns:**

["NO or VERY MINOR ethics concerns only"]

**Final Justification:**

This is a good paper, I tend to accept it.

**Limitations:**

Yes. The authors have discussed their limitations.

**Quality:**

4

**Strengths And Weaknesses:**

## Strength

1. The comparison of few-shot and many-shot settings for ICL problems is fundamental and impactful for better understanding of the transformers models.

2. The paper presents a clear and compelling dichotomy: ICL is near-optimal in the few-shot regime but becomes substantially suboptimal in the many-shot regime.

3. The paper supports its empirical findings with a solid information-theoretic analysis in Section 4. By decomposing the ICL error into Bayes risk and excess risk, and arguing that the excess risk for ICL is non-vanishing.

## Weakness

1. The primary limitation, which the authors acknowledge, is the use of a stylized setting (regression on a mixture of Fourier series). Although it's a clean setup, it raises questions about the *generalizability* of the findings to the complex, high-dimensional, and often discrete tasks that real-world LLMs handle.

2. The theoretical results hinge on Assumption 4, which posits a non-vanishing lower bound on the excess risk. While this assumption is well-supported by the paper's own experiments, it is still an assumption about the behavior of transformer-based ICL. I would like to hear more discussion about this assumption.

---

> ### Author Rebuttal · Authors · 2025-07-31
>
> We are grateful to Reviewer kMFD for the encouraging and exceptionally clear feedback. We are pleased that the reviewer found the few-shot vs. many-shot dichotomy "clear and compelling" and our analysis "solid." The questions raised are thoughtful and address the core foundations of our work as well as its broader implications.
>
> ## **W1:** “The primary limitation, which the authors acknowledge, is the use of a stylized setting (regression on a mixture of Fourier series). Although it's a clean setup, it raises questions about the generalizability of the findings to the complex, high-dimensional, and often discrete tasks that real-world LLMs handle.”
>
> This is a crucial point. While our experiments use a controlled setup, we argue that the phenomenon we identify—diminishing ICL efficiency with long context—is a fundamental issue likely to generalize to large-scale models. Our reasoning is twofold:
> - General Theoretical Foundation: Our theoretical analysis in Section 4 is formulated with a general setting. The core finding relies on Assumption 4.1, which is not tied to a specific model size or data distribution. As a result, the theoretical insights we offer remain broadly applicable, transcending the specifics of our experimental setup.
> - Alignment with Large-Scale Empirical Evidence: The practical consequence of our theoretical finding is that Assumption 4.1 is a sufficient condition for the diminishing efficiency in long context. Crucially, Assumption 4.1 generality aligns well with empirical observations from recent large-scale transformer studies. For instance, Anil et al. (2022) and Zhou et al. (2024) both report that even state-of-the-art LLMs struggle to extrapolate when test-time context length exceeds that seen during training. Given that the Bayes risk improves with more demonstrations, this means that the excess risk curve is deteriorating in the extrapolating regime.
>
> Therefore, while explicitly confirming the phenomenon in state-of-the-art LLMs remains a valuable future direction, the alignment of our results with recent large-scale studies strongly supports the practical relevance and generalizability of our findings.
>
> Anil et al. (2022). Exploring length generalization in large language models. In NeurIPS.
>
> Zhou et al. (2024). Transformers can achieve length generalization but not robustly. arXiv.
>
>
>
>
>
> ## **W2:** “The theoretical results hinge on Assumption 4, which posits a non-vanishing lower bound on the excess risk. While this assumption is well-supported by the paper's own experiments, it is still an assumption about the behavior of transformer-based ICL. I would like to hear more discussion about this assumption.”
>
> See the response to Q1.
>
>
> ## **Q1:** "The core of the theoretical argument rests on the non-vanishing excess risk. Your experiments (Fig 3b, Fig 4) convincingly show this behavior. Do you have a hypothesis for why the Transformer architecture, trained with this objective, leads to a non-vanishing excess risk? Is it related to the fixed-capacity nature of the attention mechanism or something else?"
>
>
> This is an excellent question that prompted us to conduct new experiments to strengthen our claims. Our initial focus on the Transformer was due to its dominance in ICL research and state-of-the-art large language models. However, the core of our theoretical analysis is not specific to the attention mechanism but rather to the general properties of sequence models, as long as it satisfies the conditions of our analysis (e.g., Assumption 4.1).
>
>
> Motivated by the reviewer's suggestion, we ran additional experiments with LSTM and Mamba architectures. The results are highly informative: The Transformer consistently outperformed both LSTM and Mamba in overall ICL performance, aligning with recent comparative studies (Akyürek et al., 2024; Lee et al., 2024). Crucially, the central phenomenon of diminishing sample efficiency in long context persisted across all three architectures. This finding is also consistent with recent work analyzing the length extrapolation limits of Mamba (Ben-Kish et al., 2025). These new findings strongly reinforce our central claim: the diminishing efficiency of ICL in long context is a general property of the ICL paradigm in sequence models having a non-vanishing excess risk, not an artifact of the Transformer architecture. This underscores the broad applicability of our theoretical framework. We will add a discussion of these results to the revised manuscript.
>
>
> Akyürek et al. (2024). In-context language learning: Architectures and algorithms. arXiv
>
> Lee et al.  (2024). Is attention required for icl? exploring the relationship between model architecture and in-context learning ability. ICLR
>
> Ben-Kish et al. (2025) Decimamba: Exploring the length extrapolation potential of mamba. In ICLR.
>
>
>
>
>
>
> ## **Q2:** "You conclude by motivating a new generation of 'on-the-fly' adaptive methods without the diminishing efficiency. Could you elaborate on what such methods might look like?"
>
>
> Thank you for this excellent forward-looking question. Our paper's theoretical analysis provides a clear target for designing the next generation of methods: they must directly address the non-vanishing excess risk, which we identify as the root cause of diminishing ICL efficiency and is widely observed in many transformer-based models and other sequential models.
>
> One promising direction is a method we might call Residual In-Context Learning. This approach augments the fixed base model with a lightweight, adaptive module (e.g., a linear model on the base model's features) trained at inference time. Using only the in-context examples, this module's goal is to learn the base model's residual error for the current task. Because this adaptive module is a principled learning algorithm, its error in estimating the residual would vanish as the number of demonstrations grows. Then, the final output can be obtained by correcting the original ICL prediction with the on-the-fly estimate of the excess risk, which could make the excess risk vanishing and therefore restore efficient learning behavior in long context that the base ICL model lacks.
>
> We emphasize, however, that this is a conceptual direction, and realizing such a hybrid system would require significant further investigation.

---

> > ### Comment · Reviewer_kMFD · 2025-08-05
> >
> > Thank you for your detailed rebuttal response. It resolves my questions and I keep my positive rating for this paper.

---

> > > ### Author Response · Authors · 2025-08-05
> > >
> > > We are glad to hear that our detailed response resolved your questions. Thank you again for your insightful feedback.

---

### Decision · Program_Chairs · 2025-09-17

**Decision:**

Accept (poster)

**Comment:**

This paper investigates the sample efficiency of in-context learning (ICL), aiming to determine how many demonstrations are needed to achieve a target prediction error, relative to Bayes-optimal or principled learning algorithms. The central finding is clear: ICL approaches the efficiency of a Bayes-optimal estimator in few-shot settings, but its efficiency declines as the context length increases. This behavior is explained through an information-theoretic decomposition into Bayes risk and a non-vanishing excess risk.

Three reviewers recommend acceptance; one remains rejection. Reviewer concerns focus on whether the limitation is inherent to ICL versus a by-product of imperfect training and on the scope of Assumption 4.1.

In their rebuttal, the authors offer clarifications that address the two main concerns: given that practical transformers are inherently imperfect, the ICL mechanism itself leads to diminishing efficiency with longer context lengths. The authors also clarify Assumption 4.1 as model- and environment-specific and situate it alongside empirical evidence on length generalization. While some questions regarding this assumption persist, the paper presents a novel and valuable contribution, and the authors have addressed the key concerns constructively. I recommend acceptance.